# Conformational coupling of redox-driven Na⁺-translocation in *Vibrio cholerae* NADH:quinone oxidoreductase

Jann-Louis Hau [1], Susann Kaltwasser [2], Valentin Muras[1], Marco S. Casutt[1], Georg Vohl[1], Björn Claußen[1], Wojtek Steffen[1], Alexander Leitner [3], Eckhard Bill[4], George E. Cutsail III[4], Serena DeBeer [4], Janet Vonck [5]✉, Julia Steuber [1]✉ & Günter Fritz [1]✉

In the respiratory chain, NADH oxidation is coupled to ion translocation across the membrane to build up an electrochemical gradient. In the human pathogen *Vibrio cholerae*, the sodium-pumping NADH:quinone oxidoreductase (Na⁺-NQR) generates a sodium gradient by a so far unknown mechanism. Here we show that ion pumping in Na⁺-NQR is driven by large conformational changes coupling electron transfer to ion translocation. We have determined a series of cryo-EM and X-ray structures of the Na⁺-NQR that represent snapshots of the catalytic cycle. The six subunits NqrA, B, C, D, E, and F of Na⁺-NQR harbor a unique set of cofactors that shuttle the electrons from NADH twice across the membrane to quinone. The redox state of a unique intramembranous [2Fe-2S] cluster orchestrates the movements of subunit NqrC, which acts as an electron transfer switch. We propose that this switching movement controls the release of Na⁺ from a binding site localized in subunit NqrB.

In respiratory chains of mitochondria and bacteria, the oxidation of NADH and reduction of ubiquinone are coupled to ion translocation and build-up of an electrochemical gradient[1]. This reaction is catalyzed by complex I or the Na⁺-NQR, but both membrane protein complexes have a completely different architecture. Notably, Na⁺-NQR occurs only in bacteria and is widespread among pathogens like *V. cholerae* and multidrug-resistant *Pseudomonas* and *Klebsiella* strains, making it a promising target for new antibiotics. Na⁺-NQR has a unique set of cofactors[2] (one flavin adenine dinucleotide (FAD), two covalently bound flavin mononucleotides (FMNs), one riboflavin, and two iron-sulfur centers) and couples NADH:ubiquinone oxidoreduction to the translocation of two Na⁺ ions across the cytoplasmic membrane[3]. A 3.5-Å-resolution X-ray structure of Na⁺-NQR from our group revealed the arrangement and topology of the six subunits[2]. However, the mechanism of Na⁺-NQR still

was not understood, because the structure provided just a snapshot of only one state of the catalytic cycle. The limited resolution of this X-ray structure and recent cryo-EM structures[4] did not allow resolution of water molecules or ions, and the positions of side chains and the cofactor riboflavin were not unambiguously resolved. The high-resolution cryo-EM structures reported here reveal the exact localization and nature of all cofactors, including a [2Fe-2S] cluster in subunits NqrD and NqrE. Moreover, two sodium ions were identified in NqrB.

## Results

### Architecture of the Na⁺-NQR

We determined the structure of Na⁺-NQR in different conformations by cryo-electron microscopy (cryo-EM) and X-ray crystallography. A total of six cryo-EM structures of Na⁺-NQR in the native state and

¹Department of Cellular Microbiology, Institute of Biology, University of Hohenheim, Stuttgart, Germany. ²Central Electron Microscopy Facility, Max Planck Institute of Biophysics, Frankfurt am Main, Germany. ³Department of Biology, Institute of Molecular Systems Biology, ETH Zürich, Zürich, Switzerland. ⁴Max Planck Institute for Chemical Energy Conversion, Mülheim an der Ruhr, Germany. ⁵Department of Structural Biology, Max Planck Institute of Biophysics, Frankfurt am Main, Germany. ✉e-mail: janet.vonck@biophys.mpg.de; julia.steuber@uni-hohenheim.de; guenter.fritz@uni-hohenheim.de

in complex with ubiquinone-1 (UQ-1), ubiquinone-2 (UQ-2), NADH and ubiquinone-2, the inhibitor 2-decyl-4-quinazolinyl amine (DQA), or the inhibitor 2-heptyl-4-hydroxyquinoline n-oxide (HQNO) were determined at resolutions of 2.1 to 3.2 Å (Table 1 and Extended Data Figs. 1 and 2). Moreover, we refined the X-ray structure with the new model and determined the X-ray structure of the DQA-treated state. These structures are complemented by high-resolution structures of the soluble domain of NqrF with its substrate, NADH, bound (Table 2).

Na$^+$-NQR is composed of three integral membrane subunits, NqrB, NqrD, and NqrE, and three hydrophilic subunits, NqrA, NqrC, and NqrF. NqrA and NqrF protrude into the cytoplasm, whereas NqrC is located in the periplasm (Fig. 1). NqrC and NqrF are anchored in the membrane by a single transmembrane helix, whereas NqrA, which lacks transmembrane helices, is tightly bound to NqrB. Several lipid and detergent molecules bound to NqrB, NqrD, and NqrE were resolved (Extended Data Fig. 3).

### Electron shuttling from NqrF to NqrD and NqrE

Subunit NqrF consists of two domains, an amino-terminal [2Fe-2S] ferredoxin-like and a carboxy-terminal FAD-binding domain, which is similar to ferredoxin-NADPH reductases (FNRs). Both domains are connected by a flexible linker and are anchored in the membrane by a single N-terminal transmembrane helix. The C-terminal FNR-like domain catalyzes NADH oxidation, and the electrons are transferred via the ferredoxin-like domain to the integral membrane subunits NqrD and NqrE. (NqRD-E). We have determined the cryo-EM structure of Na$^+$-NQR in the presence of NADH and UQ-2 at resolution of 2.86 Å, which was improved by density modification to 2.55 Å. The bound NADH is clearly visible in the map, and the nicotinamide is Π-stacked parallel to the isoalloxazine of the FAD for efficient hydride transfer (Extended Data Fig. 4a,b). The mode of NADH binding was further characterized by X-ray structures of the isolated FNR-like domain of NqrF, as well as of a variant of NqrF lacking the C-terminal F406, which is important for binding of NADH (Extended Data Fig. 4c,d).

In the cryo-EM and X-ray structures reported here, the ferredoxin-like domain of NqrF resides in different positions (Fig. 1 and Supplementary Fig. 1), illustrating that this domain is flexibly linked between the transmembrane helix and the FNR-like domain. This allowed us to extrapolate the conformational changes of this domain (Supplementary Fig. 1a) and to model its movement in the catalytic cycle (Supplementary Video 1). The model predicts that the [2Fe-2S] cluster of the ferredoxin-like domain gets as close as 15 Å to the [2Fe-2S] cluster localized in the membrane subunits NqrD-E. To corroborate these large conformational movements of NqrF, we performed chemical cross-linking and monitored the cross-links by mass spectrometry (Supplementary Data Sets 1 and 2).

### An intramembranous [2Fe-2S] cluster in NqrD and NqrE

The resolution of the previous X-ray structure was too low to identify the exact nature of the previously identified redox center in subunits NqrD-E. Both subunits most likely originate from a gene duplication[5] and exhibit an inverted topology[2], with four conserved Cys residues in close proximity to the center of both subunits. The anomalous difference density suggested that more than one Fe atom was present[2]. A recent cryo-EM study by Kishikawa et al.[4] reported that a [2Fe-2S] cluster might be present in NqrD-E; however, no experimental evidence was provided. We characterized this [2Fe-2S] cluster in the membrane using a comprehensive set of spectroscopic techniques. The high-resolution structures, spectroscopic data, and kinetic and analytical data show that NqrD-E coordinate a [2Fe-2S] cluster in the midst of the membrane, although this cluster had not been previously detected by spectroscopic techniques. We generated variants that lack either [2Fe-2S]$_{NqrF}$ (NqrF-C70A) or [2Fe-2S]$_{NqrD-E}$ (NqrD-C29A) (Supplementary Table 1). Wild-type Na$^+$-NQR and the variants were analyzed by circular dichroism (CD), electron paramagnetic resonance (EPR), $^{57}$Fe Mössbauer, and

Fe Kα high-energy fluorescence detected (HERFD) X-ray absorption spectroscopy (XAS). In the variant without [2Fe-2S]$_{NqrF}$, we observed the spectroscopic signature of a [2Fe-2S] cluster located in subunits NqrD-E (Fig. 2a,b). The CD spectra of this cluster showed transitions in the UV-visible region that are typical of [2Fe-2S] ferredoxins (Fig. 2c). EPR spectroscopy of the intramembranous [2Fe-2S]$_{NqrD-E}$ exhibited a prominent feature at $g = $ ~2.01 ($g$ is a component of the anisotropic g tensor), which is consistent with a radical signal; a feature at $g = $ ~1.94 was observed only at high power settings, indicative of a fast-relaxing [2Fe-2S] cluster. The EPR spectrum is adequately reproduced by simulation with two components: an isotropic radical at $g = $ ~2.01 and a dominant axial component with $g_{||} = 2.02$ and $g_\perp = 1.94$ (Fig. 2f and Supplementary Fig. 2b–e). The zero-field 80 K Mössbauer spectrum of the [2Fe-2S]$_{NqrD-E}$ cluster corroborated these results, showing an asymmetric doublet with an isomer shift of 0.3 mm s$^{-1}$ and quadrupole splitting of 0.51 mm s$^{-1}$ (Fig. 2e). EXAFS of the [2Fe-2S]$_{NqrD-E}$ cluster revealed a typical Fe-Fe distance of 2.70 Å and Fe-S distance of 2.28 Å (Fig. 2d and Supplementary Fig. 3).

### NqrC serves as a conformational redox switch

Electrons are transferred from [2Fe-2S]$_{NqrD-E}$ to FMN$_{NqrC}$, which resides in the periplasm (Fig. 1). In the previous X-ray structure, NqrC interacts with subunits NqrD-E, and the FMN$_{NqrC}$ is deeply buried in NqrD-E and in close proximity to intramembranous [2Fe-2S]$_{NqrD-E}$. The 7.2-Å distance of FMN$_{NqrC}$ to [2Fe-2S]$_{NqrD-E}$ promotes electron transfer, whereas the large distance of ~22 Å between FMN$_{NqrC}$ and FMN$_{NqrB}$, as observed in the X-ray structure, obstructs electron transfer[6] (Supplementary Table 4). We had proposed that NqrC undergoes a large conformational change to allow for electron transfer[2]. This is confirmed by the cryo-EM structures reported here and by a recent cryo-EM study by Kishikawa et al.[4], which all show NqrC residing at NqrB (Fig. 3). The position of NqrC in the cryo-EM structures versus the X-ray structure corresponds to a rotational movement of 26° and a maximum translational movement of 21.6 Å. In the cryo-EM structures, FMN$_{NqrC}$ resides in close proximity to FMN$_{NqrB}$ at a distance of 5.7 Å, compatible with very fast electron transfer (Supplementary Table 4). This conformational change of NqrC acts as a switch in Na$^+$-NQR, bridging the large distance between the redox centers in NqrD-E and NqrB. Interestingly, we observed this switching movement of NqrC in an X-ray structure when the crystal was incubated with the inhibitor DQA. The switching of NqrC was confirmed by variants that lock NqrC via engineered disulfide bonds at either NqrB or NqrD-E (Fig. 3 and Extended Data Fig. 5).

### Quinone binding recruits a detached helix in NqrB

In NqrB, the electron is transferred from FMN$_{NqrB}$ to riboflavin$_{NqrB}$. The riboflavin was precisely localized in the high-resolution cryo-EM maps and is bound by a tight hydrogen-bond network (Extended Data Fig. 6), as described by Kishikawa et al.[4]. We confirmed the binding site of riboflavin by mutating residue D346 of NqrB, which forms two hydrogen bonds to riboflavin. Mutation of D346 results in a large drop of voltage-generation activity and and impaired growth compared with that of wild-type Na$^+$-NQR (Extended Data Fig. 6). We next localized ubiquinone-1 and ubiquinone-2 in cryo-EM structures at resolutions of 2.09 Å and 2.12 Å, respectively. The quinones bind at the rim of NqrB, which is still embedded in the membrane but close to the cytoplasmic aspect of the Na$^+$-NQR (Fig. 4a,b). The distance of 11.6 Å between the quinone head group and the isoalloxazine of riboflavin is suitable for fast electron transfer.

The inhibitor HQNO binds in the same binding pocket, although in a slightly different position, and blocks the access of quinones (Fig. 4c and Extended Data Fig. 7). The binding poses of the quinones and HQNO were further corroborated in docking calculations (Extended Data Fig. 7). Interestingly, HQNO is released by *Pseudomonas aeruginosa*[7], and Na$^+$-NQR in *P. aeruginosa* tolerates 16-fold-higher levels of HQNO than does Na$^+$-NQR in *V. cholerae*[8]. In sequence alignments, we identified

**Table 1 | Cryo-EM data collection, refinement and validation statistics**

| | NQR native | NQR–NADH–UQ-2 | NQR-UQ-2 | NQR–UQ-1 | NQR–DQA | NQR–HQNO |
|---|---|---|---|---|---|---|
| | EMD-15088, PDB 8A1T | EMD-15089, PDB 8A1U | EMD-15090, PDB 8A1V | EMD-15091, PDB 8A1W | EMD-15092, PDB 8A1X | EMD-15093, PDB 8A1Y |
| **Data collection and processing** | | | | | | |
| Magnification | ×105,000 | ×215,000 | ×165,000 | ×165,000 | ×105,000 | ×105,000 |
| Voltage (kV) | 300 | 300 | 300 | 300 | 300 | 300 |
| Electron exposure (e$^-$/Å$^2$) | 48 | 41 | 40 | 40 | 48 | 41 |
| Defocus range (µm) | −1.0−−2.8 | −0.8−−2.1 | −0.8−−2.1 | −0.8−−2.1 | −1.0−−2.8 | −0.8−−2.5 |
| Pixel size (Å) | 0.831 | 0.573 | 0.730 | 0.730 | 0.831 | 0.837 |
| Symmetry imposed | $C_1$ | $C_1$ | $C_1$ | $C_1$ | $C_1$ | $C_1$ |
| Initial particle images (no.) | 263,819 | 549,586 | 496,434 | 377,944 | 302,557 | 232,734 |
| Final particle images (no.) | 248,368 | 435,715 | 226,672 | 214,804 | 273,218 | 123,832 |
| Map resolution (Å) | 3.34 | 2.86 | 2.73 | 2.56 | 3.20 | 3.30 |
| FSC threshold | 0.143 | 0.143 | 0.143 | 0.143 | 0.143 | 0.143 |
| Map resolution after density modification (Å) | 2.68 | 2.55 | 2.12 | 2.09 | 2.59 | 3.19 |
| Map resolution range (Å) | 3.3–6 | 2.7–5 | 2.5–5 | 2.3–5 | 3.1–6 | 2.7–5 |
| Applied *B*-factor (Å$^2$) | −132 | −76 | −57 | −53 | −102 | −102 |
| **Refinement** | | | | | | |
| Initial model used (PDB code) | 4PV6 | 4PV6 | 4PV6 | 4PV6 | 4PV6 | 4PV6 |
| Model resolution (Å) | 2.9/3.4 | 2.5/2.8 | 2.1/2.4 | 1.9/2.5 | 3.1/3.5 | 3.3/3.6 |
| FSC threshold | 0.143/0.5 | 0.14/0.5 | 0.143/0.5 | 0.143/0.5 | 0.143/0.5 | 0.143/0.5 |
| Model composition | | | | | | |
| Non-hydrogen atoms | 29,896 | 30,325 | 30,261 | 30,683 | 29,843 | 30,495 |
| Protein residues | 1,889 | 1,898 | 1,892 | 1,908 | 1,893 | 1,920 |
| Ligands | 2 FMN, 2 FES, 1 RBF, 3 LMT, 3 3PE, 1 FAD, 1 NA, 1K | 2 FMN, 2 FES, 1 RBF, 4 LMT, 3 3PE, 1 FAD, 1 UQ-2, 1 NAI, 1 NA, 1K | 2 FMN, 2 FES, 1 RBF, 3 LMT, 2 3PE, 1 FAD, 2 NA, 1 UQ-2 | 2 FMN, 2 FES, 1 RBF, 3 LMT, 4 3PE, 1 FAD, 2 NA, 1 UQ-1 | 2 FMN, 2 FES, 1 RBF, 3 LMT, 2 3PE, 1 FAD | 2 FMN, 2 FES, 1 RBF, 4 LMT, 3 3PE, 1 FAD, 1 HQO |
| Water | 0 | 58 | 386 | 289 | 0 | 0 |
| *B* factors (Å$^2$) | | | | | | |
| Protein | 76.6 | 39.0 | 18.2 | 16.4 | 57.3 | 128.8 |
| Ligands | 70.1 | 35.1 | 22.1 | 19.6 | 66.8 | 125.1 |
| Water | - | 20.9 | 12.4 | 17.0 | - | - |
| R.m.s. deviations | | | | | | |
| Bond lengths (Å) | 0.002 | 0.002 | 0.002 | 0.002 | 0.003 | 0.002 |
| Bond angles (°) | 0.421 | 0.446 | 0.441 | 0.453 | 0.500 | 0.404 |
| **Validation** | | | | | | |
| Molprobity score | 1.27 | 1.12 | 1.10 | 1.09 | 1.31 | 1.18 |
| Clashscore | 4.58 | 3.24 | 3.08 | 2.99 | 5.70 | 3.87 |
| Poor rotamers (%) | 0.00 | 0.39 | 0.78 | 0.26 | 0.06 | 0.06 |
| Ramachandran plot | | | | | | |
| Favored (%) | 97.82 | 98.20 | 98.30 | 98.89 | 98.03 | 97.96 |
| Allowed (%) | 2.18 | 1.80 | 1.70 | 1.11 | 1.97 | 2.04 |
| Disallowed (%) | 0.00 | 0.00 | 0.00 | 0.00 | 0.00 | 0.00 |

that L33 of NqrB in *V. cholerae* is replaced by a phenylalanine in NqrB of *P. aeruginosa*. We generated a model of NqrB of *P. aeruginosa* that predicts that HQNO binding is indeed hampered by the bulky side chain of phenylalanine, whereas quinone binding is not affected.

**Structures represent snapshots of the catalytic cycle**

The different conformations observed in the cryo-EM and X-ray structures are assigned to various states of Na$^+$-NQR during the catalytic cycle. The cryo-EM structures of Na$^+$-NQR either isolated or bound to UQ-1 or UQ-2 represent the resting or oxidized state of the respiratory enzyme, in which NqrC resides at NqrB and NqrF flexibly adopts different conformations resulting in weak density for this subunit (Extended Data Fig. 2). Structural information on reduced Na$^+$-NQR was obtained by a cryo-EM sample with a fivefold excess of NADH over UQ-2, which presumably results in the reduction of most redox cofactors. Nevertheless, in this structure, NqrC resides at NqrB, like in the oxidized state. Minor changes are observed for NqrF,

**Table 2 | Statistics for X-ray data collection, refinement, and validation**

| | Na$^+$-NQR native 1 | Na$^+$-NQR native 2 | DQA-treated Na$^+$-NQR | NqrF$_{(129-408)}$+NADH | NqrF$_{(129-408)}$-F406A | NqrF$_{(129-408)}$-F406A+NADH |
|---|---|---|---|---|---|---|
| **Data collection** | | | | | | |
| **Space group** | $P2_1$ | $P2_1$ | $P2_1$ | $P2_12_12_1$ | $P2_12_12_1$ | $P2_12_12_1$ |
| **Cell dimensions** | | | | | | |
| **a, b, c (Å)** | 95.2, 143.3, 104.3 | 93.7, 142.9, 103.4 | 89.3, 142.1, 105.9 | 27.2, 89.3, 95.2 | 73.5, 89.5, 89.65 | 2.8, 90.5 94.5 |
| **α, β, γ (°)** | 90, 110.9, 90 | 90, 109.9, 90 | 90, 109.8, 90 | 90, 90, 90 | 90, 90, 90 | 90, 90, 90 |
| **Resolution (Å)\*** | 48.7–3.5 (3.6–3.5) | 48.6–3.4 (3.5–3.4) | 48.1–3.1 (3.2–3.1) | 48.3–1.5 (1.55–1.50) | 44.8–1.55 (1.60–1.55) | 45.2–1.65 (1.7–1.65) |
| **$R_{merge}$\*** | 0.084 (3.27) | 0.266 (7.80) | 0.099 (2.31) | 0.069 (1.04) | 0.085 (1.77) | 0.154 (2.61) |
| **$I/\sigma(I)$\*** | 10.59 (0.52) | 13.61 (0.51) | 7.36 (0.49) | 14.09 (0.70) | 12.22 (0.87) | 12.44 (0.93) |
| **Completeness (%)\*** | 99.5 (97.3) | 99.3 (99.1) | 99.6 (99.5) | 94.6 (71.5) | 99.8 (98.8) | 97.9 (96.5) |
| **Redundancy** | 6.8 (7.0) | 62.7 (63.3) | 3.4 (3.4) | 4.2 (2.2) | 7.7 (7.2) | 10.6 (10.9) |
| **Refinement** | | | | | | |
| **Resolution (Å)\*** | 48.7–3.5 (3.6–3.5) | 48.6–3.4 (3.5–3.4) | 48.1–3.1 (3.2–3.1) | 48.3–1.5 (1.55–1.50) | 44.8–1.55 (1.60–1.55) | 45.2–1.65 (1.7–1.65) |
| **No. reflections\*** | 32,986 (3,201) | 35,115 (3,498) | 44,602 (4,438) | 93,846 (7,024) | 86,127 (8,435) | 74,106 (7,211) |
| **$R_{work}$ / $R_{free}$ (%)\*** | 26.6 / 29.0 (41.9 / 44.5) | 26.9 / 30.7 (38.8 / 38.4) | 24.8 / 28.7 (47.4 / 48.3) | 17.8 / 20.5 (50.0 / 50.3) | 17.4 / 20.3 (30.9 / 32.0) | 18.1 / 21.4 (30.6 / 33.2) |
| **No. atoms** | 14,344 | 14,308 | 14,408 | 5,351 | 5,056 | 5,192 |
| **Protein** | 1,490 | 1,4054 | 1,4096 | 4,538 | 4,502 | 4,515 |
| **Ligand** | 480 | 478 | 615 | 484 | 108 | 196 |
| **Water** | 0 | 0 | 5 | 499 | 446 | 481 |
| **B factors (Å)$^2$** | 176.3 | 198.3 | 136.3 | 24.5 | 29.7 | 27.9 |
| **Protein** | 176.1 | 198.4 | 136.4 | 23.3 | 29.1 | 27.0 |
| **Ligand** | 183.2 | 193.7 | 131.9 | 32.6 | 24.4 | 26.7 |
| **Water** | - | - | 110.6 | 30.0 | 37.1 | 36.5 |
| **R.m.s. deviations** | | | | | | |
| **Bond lengths (Å)** | 0.006 | 0.005 | 0.010 | 0.016 | 0.013 | 0.007 |
| **Bond angles (°)** | 1.30 | 1.18 | 1.63 | 1.37 | 1.31 | 0.96 |

\*Values in parentheses are for the highest-resolution shell.

which exhibits less flexibility and in which the ferredoxin-like domain moved 2 Å closer to the FNR-like domain of NqrF than in the oxidized state. It appears that it is not possible to induce certain conformations by a certain redox state at thermodynamic equilibrium. Strikingly, the X-ray structures revealed different conformations than did the cryo-EM structures. The most prominent difference between the X-ray and cryo-EM structures is the position of NqrC with respect to the other subunits. The X-ray structures of Na$^+$-NQR show NqrC plugged into NqrD-E (Figs. 1b and 3); however, in all structures determined by cryo-EM, NqrC resides at NqrB. This strongly suggests that the different conformation of NqrC in the X-ray structure is due to the crystallization of Na$^+$-NQR. The conformational change is not due to pH or ionic strength of the crystallization conditions, because these are similar to the buffer conditions used for the cryo-EM samples. We concluded that the packing of the Na$^+$-NQR in the crystal provides enough energy to stabilize NqrC in its position at NqrD-E. In fact, the subunit arrangement observed in the X-ray structure is largely stabilized by the packing of the molecules in the crystal. NqrC interacts in the crystal with two neighboring Na$^+$-NQR molecules, forming three salt bridges and several hydrogen bonds, thus stabilizing a conformation that represents a certain state of the catalytic cycle. Because this state is not observed at thermodynamic equilibrium, it is most likely a transient or metastable state that is trapped by the crystal packing.

The close proximity of FMN$_{NqrC}$ to [2Fe-2S]$_{NqrD-E}$, with a distance of 7.2 Å, suggests that it represents a snapshot of transmembrane electron transfer. This state is not very stable, as shown by experiments treating the protein in the crystals with the inhibitor DQA. Despite the intensive crystal contacts, incubation of the crystals with the inhibitor triggered a large conformational change of NqrC, which moved from NqrD-E towards NqrB, even within the crystals. Yet in this structure, NqrC does not reach the final relaxed position at NqrB that was observed in the cryo-EM structures, and some residual crystal contacts capture Na$^+$-NQR in a transient state.

In summary, we can distinguish three conformations of Na$^+$-NQR that reflect different states: (1) the relaxed state, as observed in the cryo-EM structures; (2) a short-lived transmembrane electron-transfer state that is shown in the X-ray structure of Na$^+$-NQR; and (3) an intermediate state between these two states, obtained after treatment of the crystals with DQA.

A systematic analysis of the structures revealed that the states differ not just in the position of NqrC, but also in the conformation of NqrD-E and in the position and interhelical angles of the transmembrane helices of NqrC and NqrF (Extended Data Fig. 8), showing that the various movements of the different subunits are tightly interconnected.

Considering the electron flow from NqrF to NqrD-E and subsequently to NqrC, the [2Fe-2S]$_{NqrD-E}$ cluster in the oxidized state must be accessible to the NqrF ferredoxin-like domain to accept electrons at the cytoplasmic side. Once the [2Fe-2S]$_{NqrD-E}$ cluster is reduced,

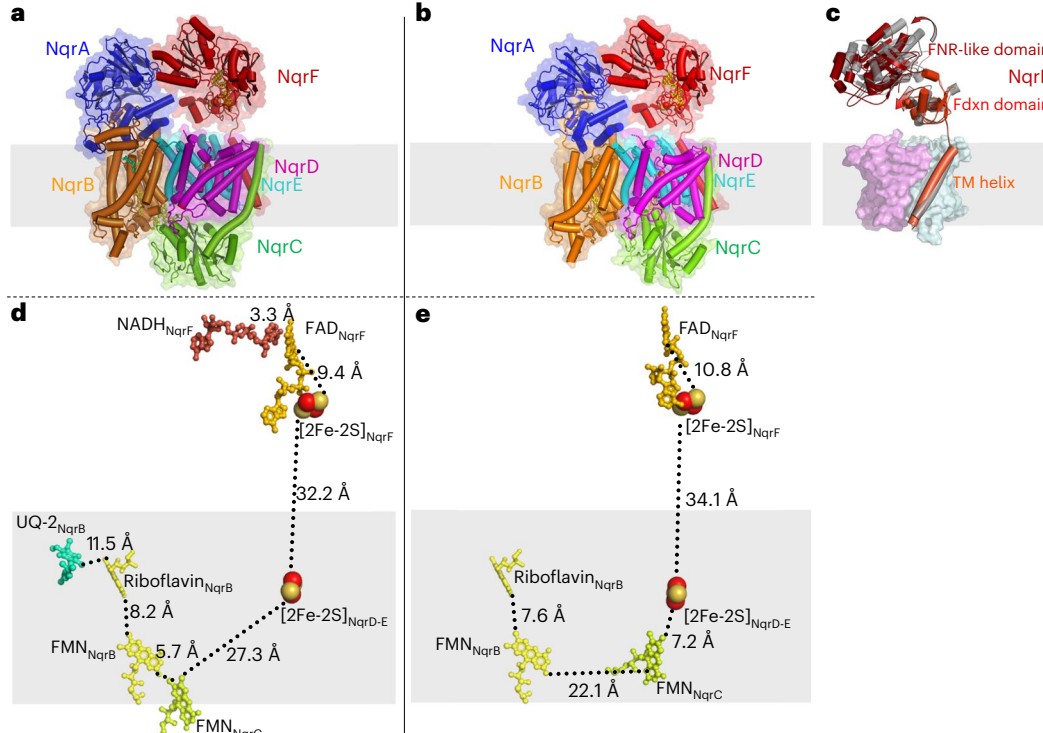

**Fig. 1 | Comparison of cryo-EM and X-ray structures of Na⁺-NQR. a**, Overall structure of Na⁺-NQR in complex with NADH and ubiquinone-2, determined by cryo-EM. **b**, Structure of Na⁺-NQR determined by X-ray crystallography. Both structures differ in the position of the periplasmic hydrophilic domain of NqrC. **c**, Structural overlay of the cryo-EM (red) and X-ray (gray) structures of NqrF bound to NADH and UQ-2. NqrD (magenta) and NqrE (cyan) are shown as surface. In comparison to the X-ray structure, the FNR-like domain of NqrF is shifted sideways, and the ferredoxin (Fdxn)-like domain is rotated towards the membrane subunits in the cryo-EM structure. **d**, Edge-to-edge distances of redox cofactors in Na⁺-NQR determined by cryo-EM. **e**, Edge-to-edge distances of redox cofactors in Na⁺-NQR determined by X-ray crystallography. Gray bars indicate the location of the membrane.

it must be accessible from the periplasmic side to transfer the electron to $FMN_{NqrC}$. Thus, we propose that the different conformations of NqrD-E are controlled by the redox state of the $[2Fe-2S]_{NqrD-E}$ cluster, and the different conformations promote either interaction of NqrC with NqrD-E at the periplasmic side or interaction of ferredoxin-like domain of NqrF with NqrD-E on the cytoplasmic aspect. Such alternating access of either the NqrF ferredoxin-like domain or NqrC to the $[2Fe-2S]_{NqrD-E}$ cluster is clearly documented in the structures (Extended Data Fig. 8 and Supplementary Videos 1 and 2). Moreover, we can assign the conformation that is open towards the cytoplasmic side to the oxidized state of $[2Fe-2S]_{NqrD-E}$, and the conformation that is open towards the periplasm to the reduced state of the $[2Fe-2S]_{NqrD-E}$ cluster. The conformational changes shifting NqrD-E from an open conformation towards the cytoplasmic side ($[2Fe-2S]_{NqrD-E}$ cluster oxidized) to the conformation that is open at the periplasmic side ($[2Fe-2S]_{NqrD-E}$ cluster reduced) involve, in particular, helix-III and helix-IV of both subunits NqrD and NqrE (Supplementary Video 2).

Helix-III and the half-helix-IV of NqrE tilt and move by 6–10 Å. NqrD and NqrE are related by an inverted topology and form a compact dimer. Thus, any conformational changes in one of these subunits will affect the other. Indeed, NqrD and NqrE are tightly coupled to each other: conformational changes of helix-III and half-helix-IV of NqrE are linked to conformational changes of helix-III and half-helix-IV of NqrD in the opposite direction.

From the observed structures, we can conclude the following: NqrD-E with an oxidized [2Fe-2S] cluster opens at the cytoplasmic aspect, and $[2Fe-2S]_{NqrD-E}$ is accessible from the cytoplasm (Extended Data Fig. 8a–c), whereas the periplasmic side is closed by an inward movement of helix-$III_{NqrD}$ and half-helix-$IV_{NqrD}$. Vice versa, if the

$[2Fe-2S]_{NqrD-E}$ cluster is reduced, NqrD-E closes at the cytoplasmic aspect by inward movements of helix-$III_{NqrE}$ and half-helix-$IV_{NqrE}$ covering the cluster, whereas the cluster becomes accessible at the periplasm by an outward movement of helix-$III_{NqrD}$ and half-helix-$IV_{NqrD}$. Only in this state it is accessible by NqrC to mediate fast electron transfer (Supplementary Video 2). In NqrD-E, these concerted movements of the helices might be triggered by the dipole moment of half-helix-I and half-helix-IV pointing to the bridging sulfurs of the $[2Fe-2S]_{NqrD-E}$ cluster. In FeS clusters the extra negative charge upon electron uptake is localized predominantly on the coordinating and bridging sulfur atoms of the cluster[9]. This additional negative charge might attract the positive dipole of one half-helix and repulse the negative dipole of the second half-helix, promoting the reorientation of the helices.

Moreover, additional interactions within the subunits can stabilize these different conformations. Notably, the conformation of NqrD-E with the closed cytoplasmic side is stabilized by a salt bridge formed between the side chains of E95 in NqrE and R71 in NqrD, whereas, in the open conformation, the residues reside at a distance of 13 Å (Extended Data Fig. 8d). As shown previously, mutated NqrE-E95 abrogates the Na⁺-pumping activity of Na⁺-NQR[10], indicating that the stabilization of this conformation is critical. Furthermore, the changes in NqrD-E going from a conformation with an oxidized $[2Fe-2S]_{NqrD-E}$ cluster to a conformation with a reduced $[2Fe-2S]_{NqrD-E}$ cluster cause a tilting of the transmembrane helices of NqrC and NqrF by 12° and 7°, respectively, as well as a shift of approximately 4 Å towards the cytoplasmic aspect of the membrane. These movements exert a pulling force on the soluble domain of NqrC towards NqrD-E and a pushing force on the ferredoxin-like domain of NqrF away from the membrane (Supplementary Video 1).

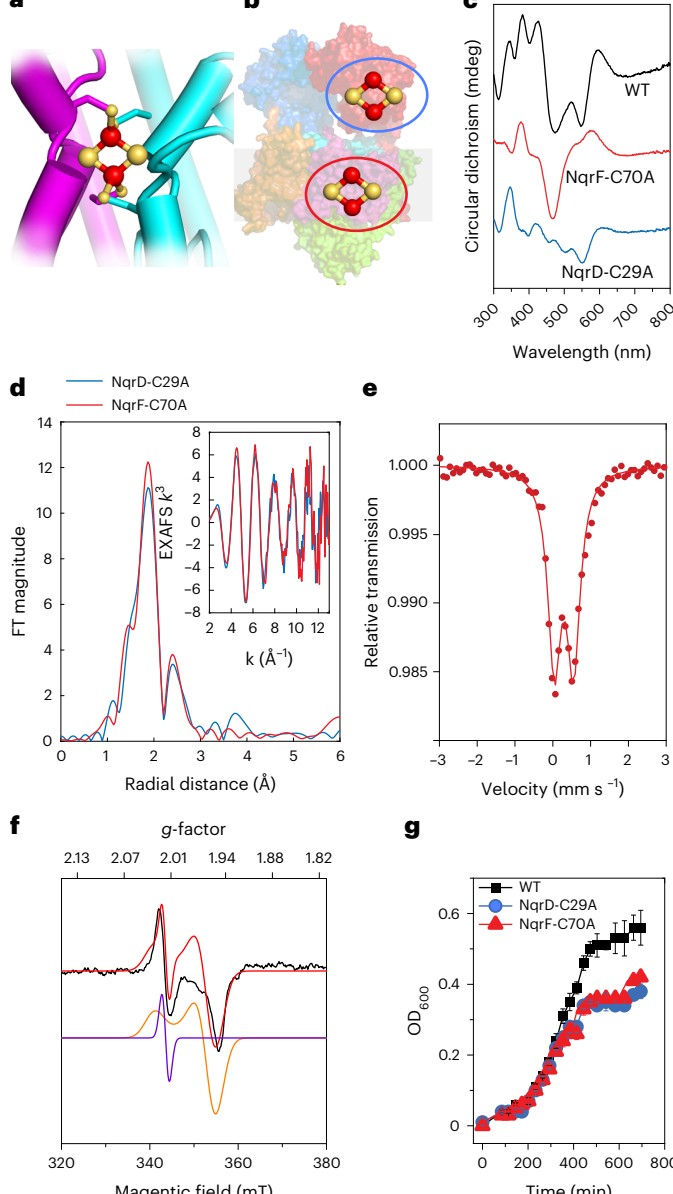

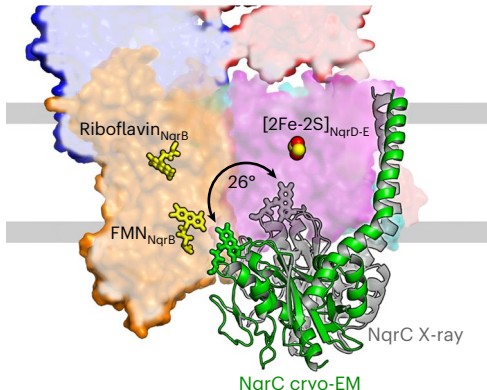

**Fig. 3 | NqrC acts as a conformational electron transfer switch.** Structural alignment of Na⁺-NQR in two states, highlighting the two conformations of NqrC trapped in cryo-EM and X-ray structures. NqrC located at NqrB, as found in the cryo-EM structure, is depicted in green. NqrC located at NqrD-E, as detected in the X-ray structure, is shown in gray. The membrane plane is indicated by gray bars. The movement of NqrC corresponds to a 26-degree rotation of the hydrophilic domain.

**Fig. 2 | NqrD-E harbors a [2Fe-2S] cluster that catalyzes transmembrane electron transfer. a**, Close-up view of subunits NqrD-E coordinating a [2Fe-2S] cluster, with cysteines originating from both subunits. **b**, Localization of the [2Fe-2S] clusters in cytoplasmic NqrF and in membrane subunits NqrD-E. The blue circle indicates the [2Fe-2S]$_{NqrF}$ cluster; the red circle indicates the [2Fe-2S]$_{NqrD-E}$ cluster. **c**, CD spectra of wild-type (WT) Na⁺-NQR; NqrF-C70A, which is devoid of [2Fe-2S]$_{NqrF}$; and NqrD-C29A, which is devoid of [2Fe-2S]$_{NqrD-E}$. Spectra of WT Na⁺-NQR (black trace), the [2Fe-2S]$_{NqrD-E}$ cluster (red trace) in NqrF-C70A, and the [2Fe-2S]$_{NqrF}$ cluster (blue trace) in NqrD-C29A are shown. **d**, $k^3$-weighted EXAFS and Fourier transform spectra of cluster [2Fe-2S]$_{NqrD-E}$ (red) in NqrF-C70A, and cluster [2Fe-2S]$_{NqrF}$ (blue) in NqrD-C29A. **e**, ⁵⁷Fe Mössbauer spectrum of the [2Fe-2S]$_{NqrD-E}$ cluster in NqrF-C70A. **f**, 10 K X-band EPR spectrum of Na⁺-NQR variant NqrF-C70A showing [2Fe-2S]$_{NqrD-E}$ cluster, simulated with two components: a [2Fe-2S] cluster (orange; $g_∥$ = 2.02, $g_⊥$ = 1.94, line width = 50 G) and a radical signal (purple; $g_{iso}$ = 2.01, line width = 20 G, 4% relative weight). **g**, Growth curves of *V. cholerae* expressing WT Na⁺-NQR (black) or Na⁺-NQR variants containing cluster [2Fe-2S]$_{NqrD-E}$ (red) in variant NqrF-C70A or cluster [2Fe-2S]$_{NqrF}$ (blue) in variant NqrD-C29A. OD$_{600}$, optical density at 600 nm. Growth data are shown as mean ± s.d., $n$ = 3 biologically independent samples.

## Na⁺ sites in NqrB

NqrB closely resembles the urea transporter[11] and the ammonium transporter[12,13], suggesting that it represents the Na⁺-translocating module

of Na⁺-NQR[2]. We observed two water-filled half-channels in the cytoplasmic and in the periplasmic half of NqrB, respectively. Of note, the access to the periplasmic half-channel to the solvent is blocked in the cryo-EM structures by NqrC. One cannot discriminate between water and ions by the intensity of the cryo-EM map; however, ion positions can be identified by the number and geometry of interacting atoms in the coordination sphere[14].

Two positions, Na-1 and Na-2 (Fig. 5a,b), have been identified as ion-binding sites, with a coordination sphere of backbone carbonyls and water molecules that is typical of alkali metal ions[15]. The map density, the average bond distances of 2.3–2.5 Å, and the coordination number of 4–5 are in excellent agreement with Na⁺. Both sites, Na-1 and Na-2, are located close to the periplasmic aspect. Na-1 is buried within NqrB, but Na-2 is at the surface of NqrB and located in close proximity to the region binding NqrC. Site Na-2, but not site Na-1, can accommodate the larger monovalent cations Rb⁺ and Cs⁺, indicating that Na-2 can bind K⁺ as well (Extended Data Fig. 9). Interestingly, K⁺ activates Na⁺-NQR[16,17]. This activation might be due to structural changes upon K⁺ binding to this site modulating the interaction of NqrC with NqrB.

## Na⁺ translocation pathway

Analysis of the high-resolution structure with the program HOLE2 (ref. 18) confirmed the presence of a putative Na⁺ translocation pathway (Fig. 5) in NqrB[2]. The pathway is lined mainly by backbone carbonyls involving L53, M57, V60, V64, A67, V161, I165, P269, G272, E274, G334, and G335 and the side chains of D52 and E157 at the cytoplasmic entry site, as well as E273 at the periplasmic exit. All three acidic residues are strictly conserved in NqrBs in different species. The proposed Na⁺ pathway includes also the site Na-1 and is remarkably narrowed halfway through the membrane by residues F342 and, in particular, F338 (Fig. 5d). Both phenylalanine residues form a putative gate between both half-channels. Interestingly, alteration of either F338 or F342 results in 30% lower Na⁺-dependent voltage formation, presumably caused by a backflow of Na⁺ during translocation because of an incomplete closure of the gate (Extended Data Fig. 10). The results corroborate that F338 and F342 in NqrB are critically involved in Na⁺ translocation.

## Discussion

An important feature of a redox-driven ion pump is the strict coupling between electron transfer and ion transport to allow for

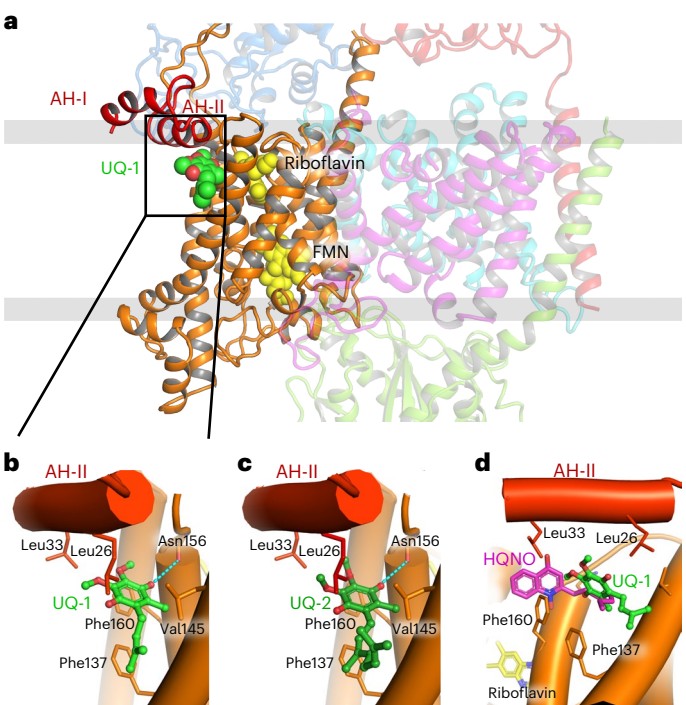

**Fig. 4 | Ubiquinone and inhibitor HQNO share the same binding site in NqrB.** **a**, Ubiquinone-1 (UQ-1, green spheres) binds to the rim of NqrB with the head group close to the cytoplasmic aspect of the membrane. **b,c**, Interactions of UQ-1 (**b**) and UQ-2 (**c**) with NqrB. The isoprenoid and the methyl group are in hydrophobic contact with residues F159, G141, and L138, whereas the O4 can form a hydrogen bond to a backbone carbonyl of N156. Replacement of G141 with any other amino acid residue would sterically block the interaction of quinones with NqrB. Two amphiphilic helices, AH-I and AH-II, at the N terminus of NqrB close upon UQ binding and contribute to the mainly hydrophobic interaction of UQ-1 or UQ-2 with NqrB. The methoxy groups of the quinone head group are in hydrophobic contact with the side chains of L26, A29, and L33 of helix AH-II. These interactions are critical and stabilize the bound cofactor in this position in an induced-fit binding mode. **d**, Structural alignment of Na$^+$-NQR in complex with either UQ-1 or with HQNO. The binding sites for both molecules overlap, but head groups and tails localize in different regions of the binding site. The quinolone head group of HQNO interacts with the side chain of F160 and L33. The alkyl chain of HQNO covers the space of the head group of quinones and follows the course of the isoprenoid tail. Like the quinones, HQNO recruits the N-terminal amphiphilic helices to form a high-affinity binding site.

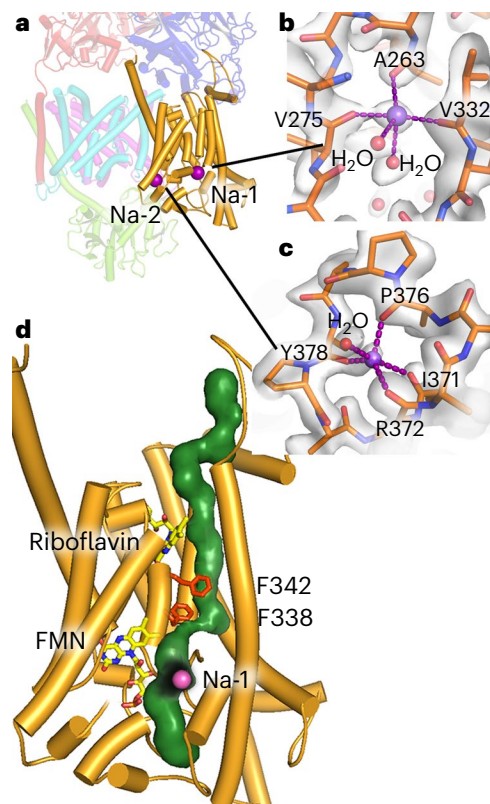

**Fig. 5 | Na$^+$ binding site and Na$^+$-translocation pathway in NqrB. a**, Two ion-binding sites were identified in NqrB. Both sites are located close to the periplasmic aspect of NqrB. **b**, Site Na-1 is buried in NqrB, and the Na$^+$ ion is coordinated by three backbone carbonyl oxygen atoms and two water molecules. **c**, Site Na-2 is exposed to the solvent, with a coordination sphere of four backbone carbonyl oxygens and one water molecule. **d**, A Na$^+$-translocation pathway through NqrB is predicted, including the position of Na-1. The translocation pathway is constricted close to residues F338 and F342.

vectorial transport against an electrochemical gradient. Typical electron-transfer rates within multi-center redox proteins are in the range of $10^6$–$10^7$ s$^{-1}$, corresponding to an edge-to-edge distance of the redox cofactors of approximately 14 Å[6]. In stark contrast, ion pumps like Na$^+$,K$^+$-ATPase operate at rates of 50–70 s$^{-1}$ (refs. [19,20]). Similarly, Na$^+$-NQR exhibits a Na$^+$-pumping rate of 56 s$^{-1}$ (ref. [21]). Thus, given suitable distances between the redox cofactors, electron transfer is several orders of magnitudes faster than ion translocation. If all six redox cofactors would reside in short distance from each other suited for fast electron transfer, the ion-translocation process would be too slow to be connected to several electron-transfer steps; that is, the energy of the different redox steps would deflagrate. To couple electron transfer to the Na$^+$-pumping process, the system has to decelerate electron transfer at certain steps to allow ion translocation to proceed. Such slow phases are intimately linked to states of Na$^+$-NQR, in which we observe large distances between the redox centers; that is, in order to decelerate electron transfer, the distance between the redox centers has to increase. Once ion translocation has occurred, conformational changes will again

move redox centers within the electron-transfer distance for fast electron transfer.

Such movements are evident when comparing the distances between FMN$_{NqrC}$ and [2Fe-2S]$_{NqrD-E}$ in the cryo-EM structure versus the X-ray structure. In the cryo-EM structure, FMN$_{NqrC}$ resides at a distance of 27.3 Å from [2Fe-2S]$_{NqrD-E}$ (Fig. 1d). This distance is shortened to 7.2 Å in the X-ray structure (Fig. 1e). These distances correspond to calculated electron-transfer rates of $1.5 \times 10^{-5}$ s$^{-1}$ for the large distance and $1.3 \times 10^7$ s$^{-1}$ for the short distance, respectively, which differ by a factor of ~$10^{12}$ (Supplementary Table 4). Consequently, in a multistep process, there must be alternating steps of 'fast electron transfer' and 'slow conformational changes' coupled to 'slow ion translocation.' The data presented here further strengthen and develop our previous model of coupling between electron transfer and Na$^+$ translocation in Na$^+$-NQR[2]. We propose a catalytic cycle as described in the following text (Fig. 6 and Supplementary Video 1). Na$^+$-NQR in the oxidized state has one Na$^+$ bound in NqrB close to the periplasmic side, as observed in the high-resolution cryo-EM structures of Na$^+$-NQR (for example, with UQ-1 or UQ-2). Once NADH binds to the FNR-like domain of NqrF, it is oxidized at FAD$_{NqrF}$; this is followed by fast electron transfer to [2Fe-2S]$_{NqrF}$ in the ferredoxin-like domain of NqrF. The flexibly linked ferredoxin-like domain of NqrF can approach the cytoplasmic aspect of NqrD-E and move the reduced [2Fe-2S]$_{NqrF}$ close enough to the intramembranous [2Fe-2S]$_{NqrD-E}$ that electrons can be transferred. This step might cause structural changes in NqrB that are presumably linked to the binding of Na$^+$ to the cytoplasmic half-channel in NqrB

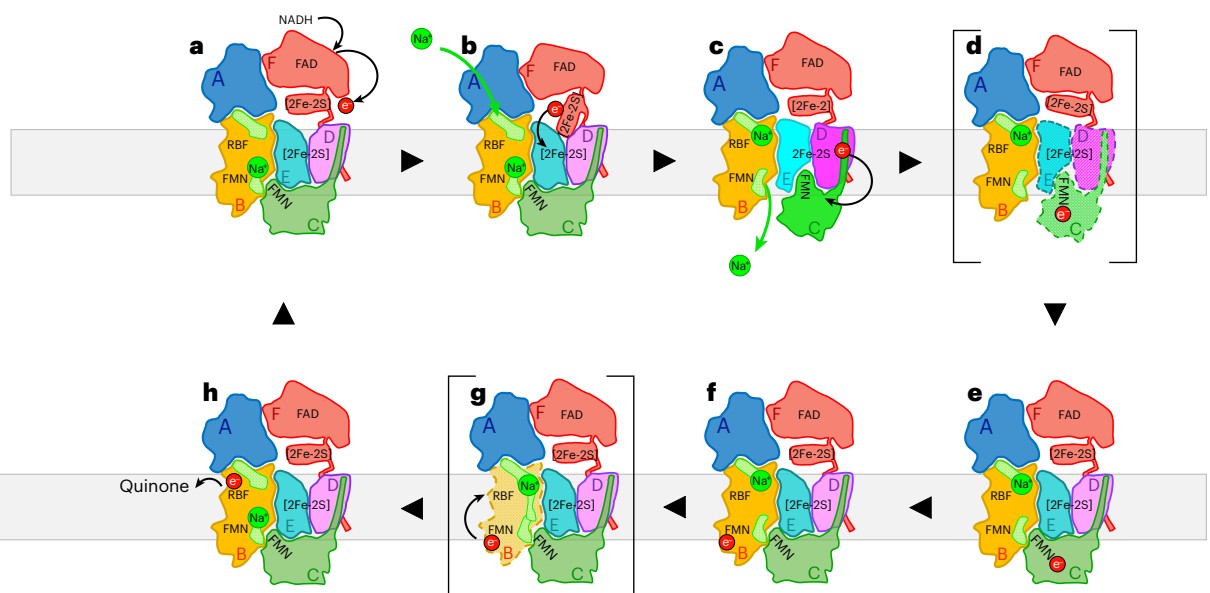

**Fig. 6 | Schematic catalytic cycle of Na⁺-NQR and mechanism of Na⁺ translocation. a**, The catalytic cycle starts with NADH becoming oxidized at NqrF. Electrons are transferred to the flexibly tethered ferredoxin-like domain of NqrF. A Na⁺ is bound in NqrB close to the periplasmic half-channel, but the release of the ion is blocked by NqrC. NqrD-E adopt a conformation that allows access of the intramembranous [2Fe-2S]$_{NqrD-E}$ cluster from the cytoplasmic side, whereas access for NqrC at the periplasmic side is blocked. **b**, The ferredoxin-like domain of NqrF can reach the [2Fe-2S]$_{NqrD-E}$ cluster and transfers an electron. This electron transfer prepares for binding of a Na⁺ to the cytoplasmic half-channel in NqrB. **c**, The reduction of [2Fe-2S]$_{NqrD-E}$ triggers a conformational switch in subunits NqrD-E, now obstructing the access of [2Fe-2S]$_{NqrD-E}$ at the cytoplasmic side but

facilitating access of NqrC at the periplasmic side. **d**, As a consequence, NqrC rotates from its position at NqrB towards NqrD-E and triggers the release of the Na⁺ bound at the periplasmic side in proximity of FMN$_{NqrB}$. The conformational change of NqrC locates FMN$_{NqrC}$ close to [2Fe-2S]$_{NqrD-E}$ resulting in fast electron transfer to the flavin. **e**, Oxidation of the [2Fe-2S]$_{NqrD-E}$ cluster will switch NqrD-E back to its previous conformation, triggering rotation of NqrC towards NqrB. **f**, Upon binding of NqrC to NqrB, the electron is transferred rapidly to FMN$_{NqrB}$. **g**, It is proposed that the reduced FMN$_{NqrB}$ triggers a transient opening between both half-channels in NqrB, enabling translocation of the Na⁺ to the periplasmic half-channel. **h**, NqrB is reoxidized by electron transfer to riboflavin$_{NqrB}$ and subsequently to ubiquinone.

(Fig. 6). The reduction of the intramembranous [2Fe-2S]$_{NqrD-E}$ cluster triggers a conformational change, closing NqrD-E at the cytoplasmic aspect and thereby preventing electron flow back to [2Fe-2S]$_{NqrF}$. Concurrently, NqrD-E open towards to the periplasm (Supplementary Video 2), allowing for access to NqrC, which moves from NqrB towards NqrD-E, bringing FMN$_{NqrC}$ close enough to the [2Fe-2S]$_{NqrD-E}$ cluster for electron transfer. We propose that this rotation of NqrC triggers the release of the Na⁺ that is bound in NqrB close to the periplasmic side (Fig. 6). Once the FMN$_{NqrC}$ is in close proximity to the [2Fe-2S]$_{NqrD-E}$ cluster, fast electron transfer to FMN$_{NqrC}$ occurs. NqrD-E with the now oxidized [2Fe-2S]$_{NqrD-E}$ favors a conformation that is closed at the periplasmic and open to the cytoplasmic aspect. This promotes flipping of NqrC with the reduced FMN$_{NqrC}$ back to NqrB. This subunit rearrangement is likely coupled to the occlusion of the Na⁺ in NqrB in the cytoplasmic half-channel. The subsequent electron-transfer steps from FMN$_{NqrC}$ to FMN$_{NqrB}$ and riboflavin$_{NqrB}$ probably trigger a transient opening between the cytoplasmic and periplasmic half-channel in NqrB. A putative gate between both half-channels is formed by residues B-F338 and B-F342, which are located on the short central helix-VIII of NqrB (Fig. 5d). The opening might be accomplished by a shift of helix-VIII along with helix-X, as proposed previously[2]. A similar movement of helices has been proposed for the homologous subunit RnfD in the related RNF complex[22], whereby the position of helix-X is controlled by the redox state of FMN$_{NqrB}$. This Na⁺-conducting state is most likely only short-lived and transient, and has not been observed in any of the available structures so far. Upon translocation, the Na⁺ will reside in the periplasmic half-channel of NqrB, because the exit is blocked by NqrC. Finally, the reduced riboflavin$_{NqrB}$ is recycled as an electron acceptor by oxidation through quinone.

Our results provide a detailed model of dynamic conformational coupling of electron transfer to Na⁺ translocation in Na⁺-NQR of the

human pathogen *V. cholerae*. This exemplifies how redox energy is converted into a chemiosmotic gradient by a respiratory complex. Na⁺-NQR is crucial for energy conservation in a plethora of Gram-negative pathogenic bacteria, including multidrug-resistant species like *Klebsiella* spp. and *Pseudomonas* spp. The structural information on HQNO binding forms an excellent basis for the development of new antibiotics.

## Online content

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

## Methods

### Na⁺-NQR expression and purification

Na⁺-NQR was expressed and purified as described previously[23], with minor modifications. Briefly, hexahistidine-tagged Na⁺-NQR was expressed in *V. cholerae* strain *Δnqr*, which lacks expression of the NQR complex. Membranes were isolated and solubilized with *n*-dodecyl-β-ᴅ-maltoside (DDM; Glycon). Na⁺-NQR was purified via Ni-Sepharose Fast Flow (Cytiva) utilizing the N-terminal hexahistidine tag fused to subunit NqrA. Purified Na⁺-NQR was dialyzed against 50 mM sodium phosphate, pH 8.0, 300 mM NaCl, 5% glycerol. Na⁺-NQR was concentrated by ultrafiltration (Amicon, 100-kDa cut-off) and further purified by size-exclusion chromatography using a Superdex 200 (16/60) column (GE Healthcare) equilibrated in 50 mM HEPES-NaOH, pH 8.5, 5% glycerol, 300 mM NaCl, and 0.05% DDM. If required, the hexahistidine tag was removed from NqrA by proteolysis with HRV-3C protease carrying a hexahistidine tag on ice prior to gel filtration. The digest was supplemented with 0.25 μM phenylmethylsulfonyl fluoride and passed again over Ni-Sepharose Fast Flow (Cytiva) in order to remove undigested protein, the cleaved hexahistidine tag, and HRV-3C protease.

### Production of Na⁺-NQR carrying mutations in NqrB

*V. cholerae* O395N1 *Δnqr*, which lacks the six structural nqr genes[24], was transformed with plasmids pNqr1, pNB_D346A, pNB_F342A, or pNB_F338A. Na⁺-NQR (wild-type or variants) was purified as described above. In all buffers, Na⁺ was replaced with K⁺. Na⁺-NQR in buffer (50 mM potassium phosphate, pH 8.0, 0.05% DDM, 300 mM KCl, 5% glycerol) was flash-frozen in liquid N₂ and stored at −80 °C until use.

### Crystallization, data collection and refinement

Crystals were grown and collected as described previously[2,25]. Briefly, Na⁺-NQR was concentrated to 7 mg ml⁻¹ by ultrafiltration (Amicon, 100-kDa cut-off) and crystallization was performed at 277 K by the sitting-drop vapor-diffusion method using Cryschem 24-1 SBS microplates (Hampton Research). Drops were set up by mixing 2 μl protein solution with 2 μl crystallization solution (40 mM KSCN, 21.0% PEG 2000 MME, 100 mM Tris-acetate pH 8.5, and 8% 1-propanol) and were equilibrated by vapor diffusion against 100 μl crystallization solution at 277 K. For cryoprotection, PEG 550 MME was added to the crystallization solution to final concentrations of 0, 5, 10, 15, or 20% PEG 550 MME, and the crystals were soaked in each solution for 5–10 s and flash-frozen immediately in liquid nitrogen. To map K⁺-binding sites in Na⁺-NQR, crystals were grown in the presence of 150 mM RbCl or CsCl, and datasets were recorded either at 0.83 Å or 1.70 Å, respectively. Highly redundant datasets with 7,200- or 10,000-degree total oscillation were collected to accurately determine the anomalous contribution of Rb⁺ and Cs⁺ ions. Data were collected at beamlines X06SA and X06DA at the Swiss Light Source using DA+ software[26] and at beamline P14 at the synchrotron PETRA III using MxCuBE[27]. All data were integrated with XDS[28] and scaled with XSCALE[28]. Structures were determined by molecular replacement with Phaser[29] using either the previous model of Na⁺-NQR (PDB: 4P6V) or a model from cryo-EM as a search model. The anomalous difference maps were calculated with anode (version 2013/1)[30] using the protein model from molecular replacement. All structures were built by iterative model building in Coot (version 0.9.81)[31] and refinement with Refmac5 (version 5.8.0350)[32] and phenix.refine (version 1.20.1)[33]. Restraints used during refinement for ubiquinone-1, ubiquinone-2, HQNO, and riboflavin were generated with acedrg (version 246)[34] from smiles codes; for energy minimization, Refmac5 (version 5.8.0350)[32] was used. Restraints for the covalent link between FMN and threonine were generated using acedrg (version 246)[34]. The cif file was edited manually using angles and distances obtained after energy minimization with gamess (version 18 AUG 2016 (R1))[35]. Harmonic restraints used during refinement in phenix.refine[33] and phenix.real_space_refine (version 1.20.1)[36] for the

coordination and geometry of the [2Fe-2S] clusters and coordination of the Na⁺ ions were generated by phenix.elbow (version 1.20.1)[37] and edited manually if required.

### Cryo-EM sample preparation and data acquisition

For preparation of cryo-EM samples, Na⁺-NQR was passed over a Superdex 200 (10/300) column equilibrated in 20 mM Tris, 50 mM NaCl, 0.01% DDM, pH 7.6, and eluted in the same buffer. The protein concentration was adjusted to about 4 mg ml⁻¹. Substrates UQ-1 and UQ-2 were added to a final concentration of 1 mM each. For a sample with UQ-2 and NADH, the substrates were added to final concentration of 1 mM and 5 mM, respectively. HQNO was added to a final concentration of 80 μM. DQA was added to a final concentration of 100 μM. Three microliters of sample was applied onto freshly glow-discharged (15 mA for 90 s in a PELCO easiGlow system) C-flat 1.2/1.3 400 mesh copper grids (Science Services). Samples were vitrified using a FEI Vitrobot Mark IV at 4 °C and 100% humidity. For native Na⁺-NQR or Na⁺-NQR with DQA or HQNO, cryo-EM data were collected automatically using EPU software version 2.1 (Thermo Fisher Scientific) with aberration-free image shift (AFIS) on a Titan Krios G3i (Thermo Fisher Scientific) microscope at 300 kV, Gatan BioQuantum imaging filter, and K3 direct electron detector operating in electron-counting mode. Videos were acquired at a nominal magnification of ×105,000, resulting in a pixel size of 0.831 Å. Movies were recorded for 2 s and subdivided into 40 frames. Electron flux rate was set to 14.5 e⁻ per pixel per second, resulting in an accumulated exposure of 41 e⁻ Å⁻² at the specimen. Data for NQR with UQ-1, UQ-2, and NADH plus UQ-2 were collected using EPU software version 2.1 (Thermo Fisher Scientific) on a Krios G4 with cold field emission gun (Thermo Fisher Scientific) and a Selectris X energy filter with a Falcon 4 detector (Thermo Fisher Scientific) at 300 kV. A 6–8-eV slit was used. The NADH plus UQ-2 dataset was collected at a nominal magnification of ×215,000, with a calibrated pixel size of 0.573 Å, and with UQ-1 or UQ-2 at ×165,000, and a pixel size of 0.730 Å. Data were collected in the EER format of the Falcon 4 for a total exposure of ~40 e⁻ Å⁻². For details, see Table 1.

### Cryo-EM image processing

Image processing was performed in RELION4 (refs. 38,39) unless otherwise specified. Dose-fractionated movies were motion-corrected with RELION's implementation of the MotionCor2 algorithm[40]. The EER data were divided into fractions with a dose of 1 e⁻ Å⁻². The contrast transfer function (CTF) was estimated using CTFFIND4.1 (ref. 41) inside RELION4 (ref. 38). Particles were picked using Topaz (version 0.2.4)[42]. Data sets were cleaned by 2D classification, and selected particles were subjected to ab initio reconstruction. Maps were refined in RELION4 (ref. 38) and the data were subjected to several rounds of CTF refinement and Bayesian polishing. As a final step, maps were treated by density modifications using phenix.resolve_cryo_em (version 1.20.1)[43]. Resolutions were estimated according to the Fourier shell correlation (FSC) 0.143 cut-off criterion of two independently refined half maps[39]. For details, see Table 1.

### Cryo-EM model building and refinement

The initial Na⁺-NQR model (PDB: 4P6V) was docked into the map and was further built using Coot (0.9.8.1)[31]. A high-resolution crystal structure of the NqrF FAD domain (PDB: 4U9U) and the AlphaFold2 (refs. 44,45) model of the NqrF ferredoxin domain were used to build into the weak NqrF density. For the NADH-bound Na⁺-NQR, the crystal structure of the NqrF-F406A mutant was used. The structures were fitted into the corresponding density, and the models were further build manually using maps sharpened by LocScale[46] and phenix.local_aniso_sharpen (version 1.20.1) or phenix.resolve_cryo_em (version 1.20.1)[43]. Putative ions were identified using the program WASP (version 1.0)[47] and CheckMyMetal[15,48].

## Chemical cross-linking

For cross-linking experiments, the Na$^+$-NQR complex was prepared at a concentration of 1 mg ml$^{-1}$ in 10 mM HEPES pH 8.0, 300 mM NaCl, 5% glycerol, and 0.05% DDM. Fifty-microliter aliquots (50 μg protein) were used for the experiments. Na$^+$-NQR was cross-linked in the absence and presence of the inhibitor DQA. DQA was added from a 50 mM stock in DMSO to a final concentration of 0.5 mM. To the inhibitor-free sample, 0.5 μl of DMSO was added. Samples were incubated for 45 min at room temperature before cross-linking.

Cross-linking was performed using a mixture of isotopically light ($d_0$) and heavy ($d_{12}$) di(succinimidyl)suberate (DSS-$d_0$/$d_{12}$, Creative Molecules). DSS was added to a final concentration of 1 mM from a 25 mM stock solution prepared in DMF, and samples were incubated for 45 min at 25 °C. To stop the reaction, ammonium bicarbonate was added to a final concentration of 50 mM from a 1 M stock solution prepared in water, and samples were incubated for another 20 min.

## Sample preparation for mass spectrometry

For reduction and alkylation of cysteine residues, first, tris(2-carboxyethyl)phosphine (50 mM in water) was added to a final concentration of 5 mM, and the sample was incubated for 30 min at 37 °C. Subsequently, iodoacetamide (100 mM in water) was added to a final concentration of 10 mM, and samples were incubated at room temperature for 30 min in the dark. This step was followed by the addition of 0.5 μg endoproteinase Lys-C (Wako, 1:100 enzyme-substrate ratio), and digestion was allowed to proceed for 2.5 h at 37 °C. Sample solutions were diluted to a final volume of 400 μl with 50 mM ammonium bicarbonate (which diluted DDM below the CMC), and 1 μg of trypsin (Promega, 1:50 enzyme:substrate ratio) was added. Proteolysis proceeded overnight at 37 °C.

Protein digests were purified by solid-phase extraction using 50 mg SepPak tC18 cartridges (Waters), and elution was performed with 300 μl of water/acetonitrile/formic acid (50/50/0.1, vol/vol/vol). After evaporation to dryness, samples were dissolved in SEC mobile phase (water/acetonitrile/trifluoroacetic acid (70/30/0.1, vol/vol/vol)), and peptides were separated by size-exclusion chromatography on an Äkta micro FPLC system (Cytiva) on a Superdex Peptide column (300 mm × 3.2 mm, GE) at a flow rate of 50 μl min$^{-1}$ (ref. [15]). Three fractions corresponding to the predominant elution range of cross-linked peptides were collected separately for mass spectrometry and were evaporated to dryness.

## Liquid chromatography–tandem mass spectrometry

Dried SEC fractions were dissolved in 20 μl of water/acetonitrile/formic acid (95/5/0.1, vol/vol/vol), and 4 μl was analyzed in duplicate on a liquid chromatography–mass spectrometry system consisting of an Easy nLC-1200 HPLC system and an Orbitrap Fusion Lumos mass spectrometer (both Thermo Fisher Scientific). Peptides were separated on an Acclaim PepMap RSLC C$_{18}$ column (25 cm × 75 μm, 2 μm particle size, Thermo Fisher Scientific) at a flow rate of 300 nl min$^{-1}$ and with a gradient of 89% mobile phase A/11% mobile phase B to 60% A/40% B in 60 min, where A is water/acetonitrile/formic acid (98/2/0.15, vol/vol/vol) and B is acetonitrile/water/formic acid (80/20/0.15, vol/vol/vol).

Peptides were sequenced in the data-dependent acquisition mode with precursor ion scans acquired in the orbitrap analyzer at a resolution of 120,000. The most abundant precursors with a charge state of +3 or higher were selected for sequencing by collision-induced dissociation in the linear ion trap at a normalized collision energy of 35%, and fragment ions were detected in the linear ion trap at rapid speed. The cycle time for sequencing was set to 3 s, and dynamic exclusion was enabled after one sequencing event for 30 s.

## Identification and quantification of cross-linked peptides

Cross-linked peptides were identified using xQuest (version 2.1.4)[49,50]. Tandem mass spectrometry data acquired in Thermo.raw format

were first converted into.mzXML format using msconvert (version 3.0.9134)[51] and were searched against a database containing the sequences of the Na$^+$-NQR target proteins and identified contaminants (three human keratins and two proteins from *V. cholerae*). DSS specificity was set to only lysine residues. Mass error tolerances for the *xQuest* search were set to ± 15 ppm at the MS1 level and to 0.2 Da and 0.3 Da (for common and xlink ion types, respectively) at the MS2 level.

Search results were filtered with a more stringent mass error tolerance corresponding to the actual error window (−5 to +3 ppm or less, depending on the data set), and a minimum total ion current value threshold of 0.1 was enforced. Spectra of the remaining identifications were manually checked, and only identifications with at least four bond cleavages per peptide or three consecutive bond cleavages per peptide were kept. An xQuest score threshold of 18 and 21.5 for intra- and inter-protein cross-link assignments was used in combination with the scoring scheme of ref. [52]; this typically corresponds to a false positive rate of <5% for databases of this size.

Quantification of cross-links was performed using xTract (version 1.0.2, available from https://gitlab.ethz.ch/leitner_lab/xtract)[49] using default parameters. Relative changes in the abundance of cross-linked peptide pairs are given as +DQA/−DQA ratios. Ratios were median-normalized and averaged over charge states and were determined for light and heavy DSS products.

## Docking calculations

Ligand-docking calculations were performed with PLANTS (version 1.2)[53,54], SMINA (version based on AutoDock Vina 1.1.2)[55], and VINAXB (version based on AutoDock Vina 1.1.2)[56]. For docking with PLANTS, structures were prepared with SPORES[57]. A search sphere with a radius of 10 Å was centered around the density for quinones. For docking with VINAXB and SMINA, a box of 16 Å × 18 Å × 16 Å was chosen. Docking calculations for UQ-1 with its short isoprenoid tail were performed within a sphere with a radius of 8 Å in PLANTS (version 1.2) and a 12 Å × 16 Å × 12 Å box in SMINA or VINAXB. Structures were refined using phenix.real_space_refine (version 1.20.1)[36].

## NqrF FAD domain plasmid construction

The amino acid sequence of NqrF from *V. cholerae* (residues 129–408) representing the FNR-like NADH-oxidizing domain[2] was obtained from Uniprot (ID: A5F5Y4). An N-terminal 6×His-tag and a subsequent HRV-3C cleavage site were added, and codons were optimized for expression in *Escherichia coli*. Genes coding for NqrF$_{129-408}$ and NqrF$_{129-408}$-F406A were synthesized by GeneArt (Thermo Fisher Scientific) and cloned into vector pET15b via flanking 5′ NcoI and 3′ BamHI, replacing the 6×His-tag and thrombin-cleavage site of pET15b, to generate plasmids pFNF53 and pFNF406A.

## Preparation of NqrF FAD domain and its variants

The *E. coli* Tuner (DE3) strain was transformed with plasmid pFNF53 or pFNF406A. Cells were grown in DYT medium containing 100 μg ml$^{-1}$ ampicillin at 37 °C. Expression of the FAD domain was induced with 1 mM IPTG at an OD$_{600}$ of 0.9. Cells were collected after 5 h at 30 °C and washed in 10 mM Tris-HCl, 0.3 M NaCl, 5 mM MgCl$_2$, pH 7.4. Cells were broken by one passage through an Emulsiflex C-3 cell disruptor (Avestin) at approximately 20 kpsi in the presence of 1 mM DTT and protease inhibitors (complete EDTA-free, Roche Diagnostics). Cell debris was removed by centrifugation at 20,000g for 20 min. After ultracentrifugation of the cellular extract at 150,000g (Beckman Type 70Ti), the supernatant was filtrated and loaded onto a Ni Sepharose HP column (5 ml bed volume, Cytiva) equilibrated with buffer A (20 mM Tris-HCl, 0.5 M NaCl, pH 8.0). The column was washed with 5 volumes of buffer A containing 30 mM imidazole, and histidine-tagged protein was eluted with 400 mM imidazole in buffer A. Peak fractions were combined and diluted at least 1:10 in 50 mM HEPES-NaOH, pH 7.0. The histidine tag was cleaved off by incubation for 15 h at 4 °C with PreScission protease.

Per 1 mg of protein, 6.7 μg of protease was added. By loading the digest onto the Ni$^+$ column and washing with 50 mM HEPES-NaOH, pH 7.0, FAD domain devoid of the histidine tag was obtained. The protein was loaded onto a SourceQ column (10 ml bed volume, Cytiva) equilibrated with 50 mM HEPES-NaOH, pH 7.0, and was eluted with a linear gradient from 0 to 0.4 M NaCl. Glycerol (5% by volume) was added, and proteins were frozen in liquid nitrogen.

### NqrF FAD domain crystallization, X-ray analysis, and structure determination

Prior to crystallization, buffer was exchanged to 5 mM Tris-Cl, pH 7.5, using NAP-5 desalting columns (Cytiva), and the protein was concentrated to 6 mg ml$^{-1}$ (10-kDa cut-off Amicon ultrafiltration spin columns, Millipore). Crystals were grown at 292 K using the hanging-drop vapor diffusion method by mixing 2 μl protein solution with 2 μl reservoir solution (22–27% PEG 5000 MME, 0.2 M magnesium acetate, 0.1 M sodium citrate, pH 5.0 or 5.2). The crystals were soaked for 15 to 20 min in the crystallization solution containing 0.2 M NADH (disodium salt, Carl Roth) or 0.2 M NADH with some grains of sodium dithionite, respectively. Crystals were passed through crystallization solution containing 35% PEG 5000 MME as a cryo-protectant and were immediately flash-frozen in liquid nitrogen. X-ray data were collected at 100 K using monochromatic synchrotron radiation ($\lambda = 1.0$ Å) at the beamline X06SA (Swiss Light Source, Paul Scherrer Institute). The diffraction data were processed with the program package XDS[28]. Structures were determined by molecular replacement with PHASER[29] using the structure of the NqrF FAD domain (PDB code 4U9U) as the search model. The structures were built by iterative model building in Coot (version 0.9.8.1)[31], and refinement was done using Refmac5 (version 5.8.0350)[32] and phenix.refine (version 1.20.1)[33].

### Construction of plasmids coding for Na$^+$-NQR variants devoid of FeS clusters

Plasmids were constructed by site-directed mutagenesis, using pNqr1 (ref. 23) as a template and specific primers. Point mutations were introduced into plasmid pNqr1 by a single PCR reaction (KAPAHiFi PCR Kit, Peqlab) with the corresponding forward and reverse primers. Mutated codons that introduce the altered amino acid residues in the Nqr subunits are underlined. Plasmid pNF-C70A codes for Na$^+$-NQR carrying the p.C70A substitution in NqrF. The forward and reverse primers were

5′-GTATTCGTATCTTCAGCTGCGGGTGGTGGTGGTTCATGTGG-3′ and 5′-CCACATGAACCACCACCACCCGCAGCTGAAGATACG-3′, respectively. Plasmid pND-C29A codes for Na$^+$-NQR carrying the p.C29A substitution in NqrD. The forward and reverse primers were 5′-TTCTGGGTGTGGCGCTGCACTGGC-3′ and 5′-CCAGTGCAGACGCCACACCCAGAAC-3′, respectively. Plasmid pNDE-C29A-C120A codes for Na$^+$-NQR carrying the p.C29A substitution in NqrD and the p.C120A substitution in NqrE. Substitution p.C29A was inserted with the same forward and reverse primers like in pND-C29A. For substitution p.C120A, the forward and reverse primers were 5′-GATCACAGTAAACGCGGCGATCTTCGG-3′ and 5′-GAAGATCGCCGCGTTTACTGTGATCAGC-3′, respectively. Plasmid pNE-C120S codes for Na$^+$-NQR carrying the p.C120S substitution in NqrE. The forward and reverse primers were 5′-GATCACAGTAAACTCAGCGATCTTCGGTGG-3′ and 5′-ACCGAAGATCGCTGAGTTTACTGTGATCAGC-3′, respectively.

### Production of Na$^+$-NQR for Mössbauer spectroscopy

To obtain Na$^+$-NQR enriched with $^{57}$Fe, *V. cholerae* transformed with pNqr1 (ref. 23) or pNF-C70A was grown in minimal medium, as described in ref. 2, to which 1 mM MgSO$_4$ and 1 ml of a SL-9 trace element solution were added. The SL-9 trace element solution contains 30 mM EDTA, 0.51 mM ZnSO$_4$, 0.62 mM MnCl$_2$ × 4 H$_2$O, 0.097 mM H$_3$BO$_3$, 0.8 mM CoCl$_2$ × 6 H$_2$O, 0.02 mM CuCl$_2$ × 2 H$_2$O, 0.11 mM NiCl$_2$ × 6 H$_2$O, and 0.16 mM Na$_2$MoO$_4$ × 2 H$_2$O, 0.7 mM CaCl$_2$ (ref. 58). The chemicals

were of the highest purity grade, with an iron content of ≤ 0.003%. *V. cholerae* was limited by iron under these conditions, as confirmed in comparative growth studies with or without added $^{56}$Fe (14 μM). $^{57}$Fe was purchased from Chemotrade (96.28% isotopic enrichment) as a metal powder and was dissolved in 2 ml 37% HCl at 60 °C. The $^{57}$Fe stock solution was neutralized with 33 ml of 0.72 N NaOH. The medium and all solutions were prepared with Millipore H$_2$O, which had been passed over a column loaded with 25 g Chelex-100. Vessels were rinsed several times with 0.1 mM EDTA, 1 M HCl, and Chelex-treated H$_2$O and were used exclusively for cell growth and purification of NQR enriched in $^{57}$Fe. New plasticware was used, where possible in the experiments. Twenty milliliters of an overnight culture of *V. cholerae Δnqr*[24] transformed with pNqr1 or pNF-C70A was used to inoculate 1 L of freshly prepared minimal medium with SL-9 and 14 μM $^{57}$Fe in a 2-L Erlenmeyer chicane flask (Polycarbonate, Nalgene). Cells were grown at 37 °C under shaking (180 r.p.m.). Protein expression was induced in the late exponential phase (OD$_{600}$, 0.6) by addition of 13 mM arabinose. Growth was continued for 20 h at 25 °C (140 rpm). On the next day, cells were collected and Na$^+$-NQR was purified as described[23]. The enrichment of $^{57}$Fe in Na$^+$-NQR was at least 70%, as confirmed by inductively coupled mass spectroscopy.

### $^{56}$Fe, $^{57}$Fe, S, and acid-labile sulfide content of Na$^+$-NQR

One milligram Na$^+$-NQR after size-exclusion chromatography was applied to a NAP-5 column for buffer exchange. Protein was eluted in Tris-HCl pH 8.0, 5% glycerol, 0.1% (by weight) DDM and was concentrated to approximately 2.5 μg μl$^{-1}$ (Vivaspin 4 centrifugal concentrator, 100,000 molecular weight cut-off, Sartorius Stedim). The amount of $^{56}$Fe, $^{57}$Fe, and S in Na$^+$-NQR was determined by inductively coupled plasma resonance mass spectroscopy (Spurenanalytisches Laboratorium Dr. Heinrich Baumann). From the amount of detected sulfur, the amount of Na$^+$-NQR in the sample was determined. Wild-type Na$^+$-NQR contains a total of 110 sulfurs (48× methionine, 58× cysteine and 2× [2Fe-2S]). For calculation of the S:Na$^+$-NQR ratio, molecular weights of 32.06 g mol$^{-1}$ (S) and 210.8 kDa (Na$^+$-NQR) were assumed. For determination of acid-labile sulfide, 2 nmol Na$^+$-NQR after size-exclusion chromatography was analyzed in triplicate, as described in ref. 59.

### Mössbauer spectroscopy

For Mössbauer spectroscopy, after the nickel affinity chromatography step[23], Na$^+$-NQR was concentrated to 110–120 mg ml$^{-1}$ to a final volume of 600 μl. Na$^+$-NQR was mixed with 66 mM sodium dithionite. To achieve full reduction of all cofactors, the protein was allowed to react with the reductant for at least 5 min under a constant stream of N$_2$. A 400-μl Mössbauer cap was filled with protein and was shock-frozen in liquid nitrogen. Spectra were recorded on a conventional spectrometer with alternating constant acceleration of the γ-source[60]. The minimum experimental line width was 0.24 mm s$^{-1}$ (full width at half-height). The temperature was maintained using a Mössbauer-Spectromag cryostat with a split-pair magnet system (Oxford Instruments Variox). The γ-source ($^{57}$Co/Rh, 1.8 GBq) was kept at room temperature. By using a re-entrant bore tube, the γ-source could be positioned inside the gap of the magnet coils at a position with zero field. Isomer shifts are quoted relative to iron metal at 300 K. The Mössbauer spectra recorded at zero field were fitted using the program MFIT with Lorentzian doublets[61].

### HERFD X-ray spectroscopic measurements

High-energy resolution fluorescence detected (HERFD)-X-ray absorption spectroscopy (XAS) was performed at the European Synchrotron Research Facility (ESRF) beamline ID-26 (6 GeV, 200 mA), equipped with a liquid helium cryostat and sample chamber operated at 20 K. XAS spectra detected by iron Kα X-ray emission were collected using a Si(311) double crystal monochromator upstream for energy selection that was calibrated to the first inflection point of an Fe reference foil set to an energy of 7,111.2 eV. The Johann X-ray emission spectrometer

was equipped with four Ge(440) crystal analyzers arranged in a Rowland geometry[62]. An energy-selective KETEK detector, set to the Kα emission fluorescence region of interest, was used to further increase signal to noise ratio. All EXAFS and XANES spectra were collected at the maximum of the Kα emission energy (~6,404 eV). Assessment of short XANES scans (5–60 s) was used to assess radiation damage processes and to determine maximum dwell time limit per spot. Only scans that showed no evidence of radiation damage were included in the final analysis. The total number of samples spots for sample 1 (Na⁺-NQR containing the [2Fe-2S] cluster in NqrF, but lacking the intramembranous [2Fe-2S] cluster) and sample 2 (Na⁺-NQR containing the intramembranous [2Fe-2S] cluster, but lacking the [2Fe-2S cluster] in NqrF) was 143 and 196, respectively (beam spot size: 1.2 mm (h) × 0.1 mm (v)). The raw XAS spectra were initially averaged in Matlab (version R2023a) and exported for further processing within Athena (version 0.9.26)[63]. A second-order polynomial was fit to the pre-edge region and subtracted throughout the entire EXAFS spectrum. A three-region cubic spline (with the AUTOBK function within Athena) was used to model the background function to a minimum of $k = 13.5$ Å⁻¹ for all spectra. Fourier transforms were performed over a windowed $k$ range of 2.0–13.0 Å⁻¹ and all FT spectra are presented without a phase-shift correction.

## EPR spectroscopy

Samples for X-band EPR spectroscopy were prepared in the anaerobic chamber. Chemicals, solutions, and materials were stored in the chamber for at least 16 h prior to use. Buffers were purged with N₂ for at least 2 h and then stored in the anaerobic chamber for at least 16 h prior to use. Na⁺-NQR (50 mg ml⁻¹) was passed over a NAP-5 column and eluted with anoxic buffer (10 mM HEPES pH 8.0, 5% glycerol, 300 mM NaCl) containing 0.05% (by weight) DDM. The peak fraction was collected, and 250 μl was used for X-band EPR spectroscopy. To achieve full reduction of Na⁺-NQR, the protein was allowed to react with dithionite (added as powder) for at least 5 min. Continuous-wave (CW) X-band EPR spectra were measured on a Bruker E500 spectrometer equipped with an Oxford liquid helium flow cryostat and a set temperature of 10 K, as described in ref. [64]. Spectra were collected in a dual-mode X-band resonator operated in perpendicular-mode (TE₁₀₂).

## CD spectroscopy

Magnetic circular dichroism spectra of Na⁺-NQR (10–20 mg ml⁻¹) were recorded on a Jasco J-715 magnetic circular dichroism (MCD) spectrometer equipped with a permanent 1.6 T magnet. Measurements were performed at room temperature using a 2 mm quartz cuvette at a speed of 100 nm min⁻¹. A magnetic field was applied and −MCD and +MCD spectra were recorded. These spectra were used to calculate CD spectra using the formula: +MCD + −MCD = 2 CD. Data from three scans were accumulated.

## UV/VIS spectroscopy

UV-visible spectra were recorded on a Lambda 12 UV/VIS spectrophotometer (Perkin Elmer) at 19 °C. Absorbance of Na⁺-NQR (50–100 μM) after gel filtration in 10 mM HEPES pH 8.0, 5% glycerol, 300 mM NaCl, and 0.05% DDM was measured in a 1 mm quartz cuvette.

## Construction of vectors for the expression of Na⁺-NQR variants carrying mutations in NqrB

Plasmids coding for Na⁺-NQR variants carrying substitutions in NqrB were created by site-directed mutagenesis, using pNqr1 (ref. [23]) as a template and specific primers. Point mutations were introduced into plasmid pNqr1 by a single PCR reaction (KAPAHiFiTM PCR Kit, from Peqlab) with the corresponding forward and reverse primers. Plasmid pNF-D346A codes for Na⁺-NQR carrying the p.D346A substitution in NqrB. The forward and reverse primers were 5′-GTTCTTCATGGCGACTGCGCCAGTTTCTGCGTC-3′ and 5′-GAAGGACGCAGAAACTGGCGCAGTCGCCATGAAG-3′, respectively.

Plasmid pNB_F342A codes for Na⁺-NQR carrying the p.F342A substitution in NqrB. The forward and reverse primers were 5′-CGCATTCGGTATGTTCGCTATGGCGACTGACCC-3′ and 5′-GAAACTGGGTCAGTCGCCATAGCGAACATACCG-3′, respectively.

Plasmid pNB_F338A codes for Na⁺-NQR carrying the p.F338A substitution in NqrB. The forward and reverse primers were 5′-GTATTGGGTGGTTTCGCAGCGGGTATGTTCTTC-3′ and 5′-CCATGAAGAACATACCCGCTGCGAAACCAC-3′, respectively. The amplified vectors were treated with DNAse I to degrade the template vector pNqr1, and transformed in *E. coli* XL 10-Gold (Stratagene). To exclude second-site errors introduced by amplification of other nqr genes in the course of the reaction, the plasmids were purified from *E. coli* and digested with Sal I/Mlu I to obtain 765-bp fragments, each carrying the desired substitution in nqrB. These fragments were ligated into the Sal I/Mlu I vector fragment (10,568 bp) of pNqr1, yielding vectors pNB_D346A, pNB_F342A, or pNB_F338A. The introduced substitutions were confirmed by sequencing (Macrogen). Transformation of competent cells of *V. cholerae* O395N1 Δnqr lacking the six structural nqr genes[24] was performed as described[21].

## Reconstitution of Na⁺-NQR in proteoliposomes

Proteoliposomes were formed by detergent dilution, as described previously[17,65] but with the following modifications. Na⁺-NQR (wild type or variants, each 2 mg in 0.3 mL), purified by nickel affinity chromatography[23] in 50 mM potassium phosphate, pH 8.0, 0.2% DDM, 300 mM KCl, and 5% glycerol, was added to a lipid film (40 mg l-α-phosphatidylcholine, from soybean, Type II-S, 14–23% as choline) that was dried under vacuum in a round bottle flask. The lipid film was gently solubilized, and 0.3 ml reconstitution buffer (20 mM Tris-H₂SO₄, pH 8.0, 5% glycerol, 50 mM K₂SO₄) was added. This suspension of lipids, detergent, and Na⁺-NQR was further diluted by dropwise addition of 0.6 ml reconstitution buffer (final volume, 1.0–1.2 ml). The dispersion of lipids and formation of vesicles was enhanced by brief ultrasonication[66] with a tip sonicator (MS73 tip with a diameter of 3 mm, Bandelin) in a 1.5-mL reaction tube on ice. The tip was placed 1 cm below the surface of the proteoliposome suspension. Three pulses of 20 s (amplitude, 30%) were performed, with cooling times of 60 s between the pulses. To decrease the detergent concentration below the critical micellar concentration (CMC), the suspension was diluted with a 50-fold volume of reconstitution buffer using a burette (30 drops min⁻¹). After ultracentrifugation (150,000g, 45 min, 4 °C), the pellet containing the proteoliposomes was suspended in 1 ml reconstitution buffer per 1 mg Na⁺-NQR, corresponding to 2 ml reconstitution buffer per 40 mg l-α-phosphatidylcholine.

## Enzyme kinetics

**NADH:quinone oxidoreduction.** NADH oxidation and ubiquinone-1 (UQ-1) reduction by Na⁺-NQR and its variants (0.4 μg protein) were followed (in the presence of oxygen) in triplicates in 20 mM Tris-H₂SO₄, pH 8.0, 50 mM K₂SO₄, 0.1% (wt/vol) DDM, and 50 μM UQ-1 (ref. [17]). The residual Na⁺ concentration of the assay buffer was 27 μM. NaCl was added as indicated (0–30 mM), and the reaction was started by the addition of 150 μM NADH (dipotassium salt). Formation of ubiquinol was sub-stoichiometric owing to the formation of superoxide as a reaction side product[67,68].

**Stopped-flow fast kinetics.** Fast kinetic measurements were performed using an SX20 stopped-flow spectrophotometer (Applied Photophysics) equipped with a diode array detector. Spectra were acquired at 20 °C in the range of 300 nm to 1150 nm. Wild-type Na⁺-NQR or the Na⁺-NQR variant (NqrD-C29A) lacking the intramembranous [2Fe-2S] cluster was added to 50 mM HEPES pH 8.0, 300 mM NaCl, pH 8.0, 5% (vol/vol) glycerol, and 0.05% (wt/vol) n-dodecyl β-maltoside and was mixed with 0.06 mM NADH and 1 mM ubiquinone-1 in the same buffer. The reaction was followed for 1,000 ms. Each measurement was

repeated eight times. The obtained spectra were averaged, and single wavelengths of interest were extracted using awk (https://www.gnu.org/software/gawk/gawk.html). The kinetics were analyzed with Origin (OriginLab, https://www.originlab.com/).

## Voltage formation by Na⁺-NQR

The formation of an inner positive membrane potential in proteoliposomes was followed from the difference in absorbance of oxonol VI (630 nm to 523 nm)[17]. Before the measurements, proteoliposomes (0.16 mg Na⁺-NQR in 3.2 mg lipids) were mixed with 6 µM oxonol VI, 50 µM UQ-1, and NaCl (as indicated) in assay buffer, which was identical to the reconstitution buffer. The total assay volume was 1.2 ml. After 10 s, the reaction was started by the addition of 0.2 mM NADH (dipotassium salt), and voltage formation was followed at 25 °C under stirring. Typical traces from three replicates are presented.

## Mapping of interaction domains between subunits of Na⁺-NQR

Vectors were created to express Na⁺-NQR variants carrying cysteine substitutions.

Gene fragments introducing cysteine into subunits NqrB, NqrC, and NqrD were synthesized (Life Technologies) and cloned into pNqr1 (ref. 23). Plasmid pNCD-P174C/Q100C codes for the Na⁺-NQR variant carrying the p.P174C substitution in NqrC and the p.Q100C substitution in NqrD. Plasmid pNBC-P376C/L226C codes for the Na⁺-NQR variant carrying the p.P376C substitution in subunit NqrB and the p.L226C substitution in subunit NqrC. Plasmids were transformed into *V. cholerae* O395N1 Δnqr lacking the six structural nqr genes.

## Characterization of Na⁺-NQR variants with intra- and intersubunit disulfide bridges

The Na⁺-NQR variants were purified as has been described[23] and flash-frozen in liquid nitrogen. Aliquots were thawed on ice and treated as indicated below, prior to SDS–PAGE[69] and activity measurements[17]. For SDS–PAGE, Na⁺-NQR (10 µg protein per lane) was incubated with loading buffer without reducing agent (40 mM Tris-HCl, pH 6.8, 2% SDS, 4% glycerol, 0.01% bromophenol blue) for 30 min at room temperature. To promote disulfide bond formation, Na⁺-NQR variants (each 0.7 mg protein) were incubated with glutathione disulfide (GSSG) reaction buffer (229 mM GSSG, 100 mM Tris-NaOH pH 8.0, 200 mM NaCl, 50 mM KCl, 5% glycerol, 0.05% DDM in a total volume of 350 µl) at 4 °C for 48 h. In the control reaction, GSSG was omitted. One aliquot (20 µl) was stored at −20 °C for subsequent SDS–PAGE analysis; (another 30 µl) was kept at 4 °C for subsequent determination of activity. The remaining reaction mixture (300 µl) was loaded on a NAP-5 gel filtration column (Sephadex G-25, Cytiva) equilibrated with buffer (300 mM NaCl, 5% glycerol, 0,05% DDM, 20 mM Tris-HCl pH 8.0). Na⁺-NQR variants were eluted with 800 µl buffer, and an aliquot (250 µl) was mixed with 50 µl 0.8 M DL-dithiothreitol (DTT) in 0.2 M Tris-HCl, pH 8.0. After 60 min, an aliquot of 50 µl was retrieved for SDS–PAGE analysis, and the remaining reaction mixture (250 µl) was loaded on a NAP-5 gel filtration column (Sephadex G-25, Cytiva) equilibrated with buffer. Na⁺-NQR variants were eluted with 500 µl buffer. NADH oxidation and ubiquinone-1 (UQ-1) reduction activities of as isolated, GSSG-treated, and DTT-treated Na⁺-NQR variants were followed in 20 mM Tris-H₂SO₄, pH 8.0, 50 mM Na₂SO₄, 0.1% DDM, and 50 µM UQ-1 (ref. 17).

## Reporting summary

Further information on research design is available in the Nature Portfolio Reporting Summary linked to this article.

## Data availability

Cryo-EM maps have been deposited in the Electron Microscopy Data Bank under accession numbers EMD-15088 (Na⁺-NQR_native), EMD-15091 (Na⁺-NQR_ubiquinone-1), EMD-15090 (Na⁺-NQR_ubiquinone-2), EMD-15089 (Na⁺-NQR_ubiquinone-2,NADH), EMD-15092 (Na⁺-NQR_DQA), and EMD-15093 (Na⁺-NQR_HQNO). Cryo-EM model coordinates have been deposited in the Protein Data Bank under accession numbers 8A1T (Na⁺-NQR_native), 8A1W (Na⁺-NQR_ubiquinone-1), 8A1V (Na⁺-NQR_ubiquinone-2), 8A1U (Na⁺-NQR_ubiquinone-2, NADH), 8A1X (Na⁺-NQR_DQA), and 8A1Y (Na⁺-NQR_HQNO).

X-ray structure coordinates and structure factors for Na⁺-NQR and of subunit NqrF_129–408 and NqrF-F406A_129–408 have been deposited in the Protein Data Bank. The PDB accession codes are 8ACY (Na⁺-NQR), 8ACW (Na⁺-NQR), 8AD0 (Na⁺-NQR_DQA), 8AD4 (NqrF_NADH), 8AD3 (NqrF-F406A), and 8AD5 (NqrF-F406A_NADH).

The cross-linking mass spectrometry data have been deposited to the ProteomeXchange Consortium via the PRIDE[70] partner repository with the dataset identifier PXD039289. Source data are provided with this paper.

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

## Acknowledgements

This work was supported by grant 311211092 from the Deutsche Forschungsgemeinschaft (G.F. and J.S.), and by fellowships from the Carl-Zeiss-Foundation (B.C.), the Landesgraduiertenförderung Baden-Württemberg (V.M.), and by the Max Planck Society (G.E.C., E.B., S.D.). The funders had no role in study design, data collection and analysis, decision to publish, or preparation of the manuscript. Synchrotron data were collected under proposal MX-346 at beamline P14 operated by EMBL Hamburg at the PETRA III storage ring (DESY, Hamburg, Germany) and at beamlines X06SA and X06DA at Swiss Light Source of the Paul Scherrer Institute (Villigen, Switzerland). We thank the Central Electron Microscopy Facility of the Max Planck Institute of Biophysics for excellent support. We thank J. Castillo-Hernández and Ö. Yildiz and the Max Planck Computing and Data Facility for maintaining the computational infrastructure for cryo-EM. We thank G. Burenkov at beamline P14 at PETRA III, the staff at beamlines X06SA and X06DA at Swiss Light Source, and B. Mienert at MPI-CEC for excellent support. The HERFD-XAS experiments were performed on beamline ID-26 at the European Synchrotron Radiation Facility (ESRF), Grenoble, France. We are grateful to B. Detlefs and L. Amidani at ESRF for assistance at beamline ID-26. We thank R. Aebersold, ETH Zurich, for access to laboratory infrastructure and instrumentation. We thank O. Einsle, University of Freiburg, Germany, and W. Kühlbrandt, Max Planck Institute of Biophysics, Frankfurt a.M., Germany, for critical discussions.

## Author contributions

J.S., G.F., M.S.C., W.S., V.M., and B.C. developed expression constructs. G.F., J.S., M.S.C., G.V., W.S., V.M., and B.C. developed

purification procedures; M.S.C., G.V., W.S., V.M., B.C., and J.-L.H. expressed and purified the protein. M.S.C., G.V., and G.F. performed crystallization and data collection. G.F. performed X-ray data processing. G.F. M.S.C., W.S., and J.-L.H. performed model building and refinement. V.M. prepared samples for Mössbauer, EPR, and XAS experiments. G.E.C. and S.D. designed XAS experiments. G.E.C. performed XAS experiments and analyzed data. E.B. performed EPR and Mössbauer experiments and analyzed data. G.F. performed fast kinetics measurements. A.L. designed, performed, and analyzed mass spectrometry analysis. S.K. prepared cryo-EM samples and performed data collection. J.V. processed the cryo-EM data and performed model building and refinement. G.F., J.-L.H., and J.V. prepared figures. A.L., G.E.C., J.V., J.S., and G.F. wrote the manuscript.

## Funding

## Competing interests

The authors declare no competing interests.

## Additional information

**Extended data** is available for this paper at https://doi.org/10.1038/s41594-023-01099-0.

**Correspondence and requests for materials** should be addressed to Janet Vonck, Julia Steuber or Günter Fritz.

**Peer review information** : *Nature Structural & Molecular Biology* thanks Leonid Sazanov and the other, anonymous, reviewer(s) for their contribution to the peer review of this work. Primary Handling Editors: Carolina Perdigoto and Katarzyna Ciazynska, in collaboration with the *Nature Structural & Molecular Biology* team. Peer reviewer reports are available.

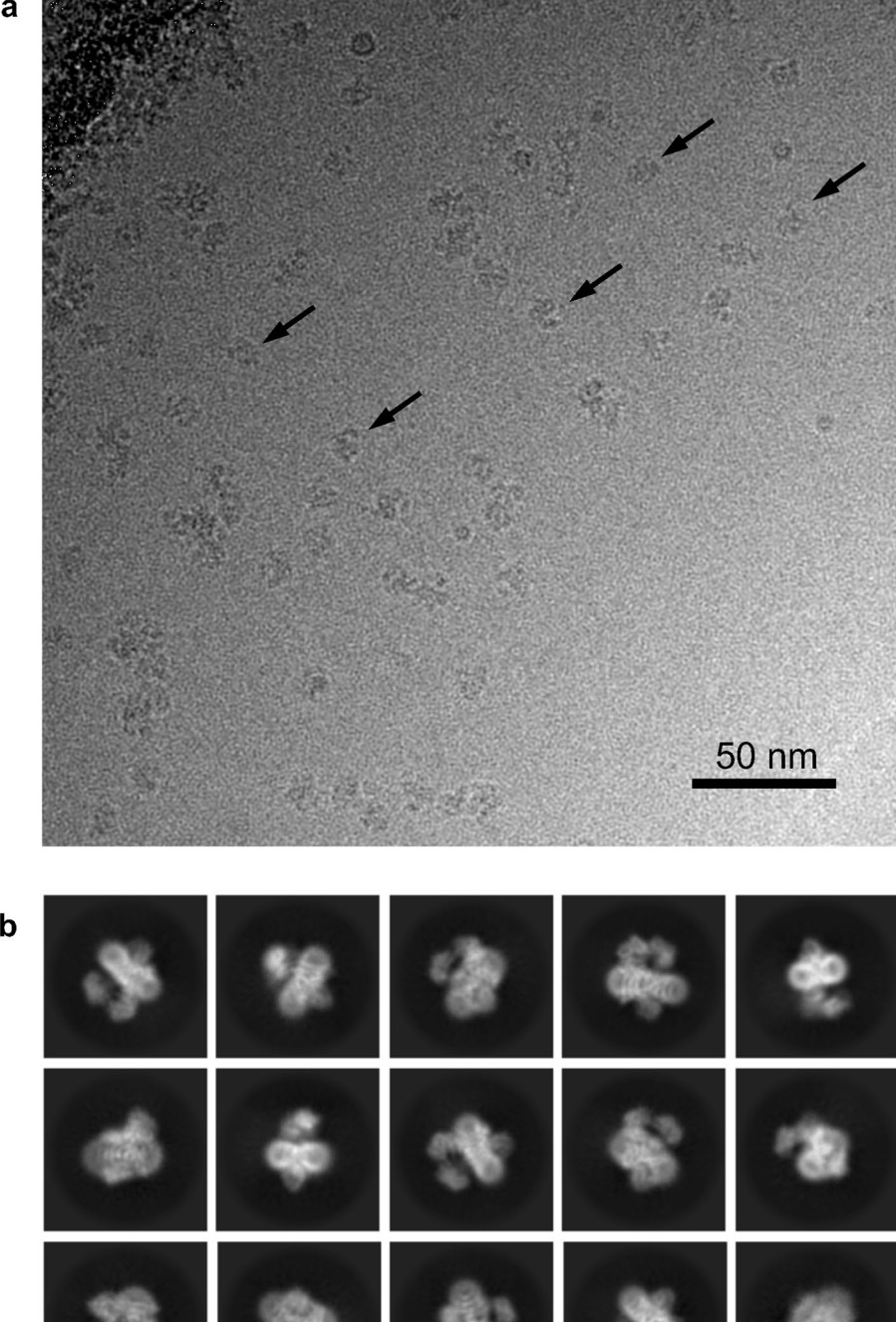

**Extended Data Fig. 1 | Micrographs and 2D classes of Na$^+$-NQR. a**, Representative micrograph of Na$^+$-NQR with UQ-1. **b**, Representative 2D class averages.

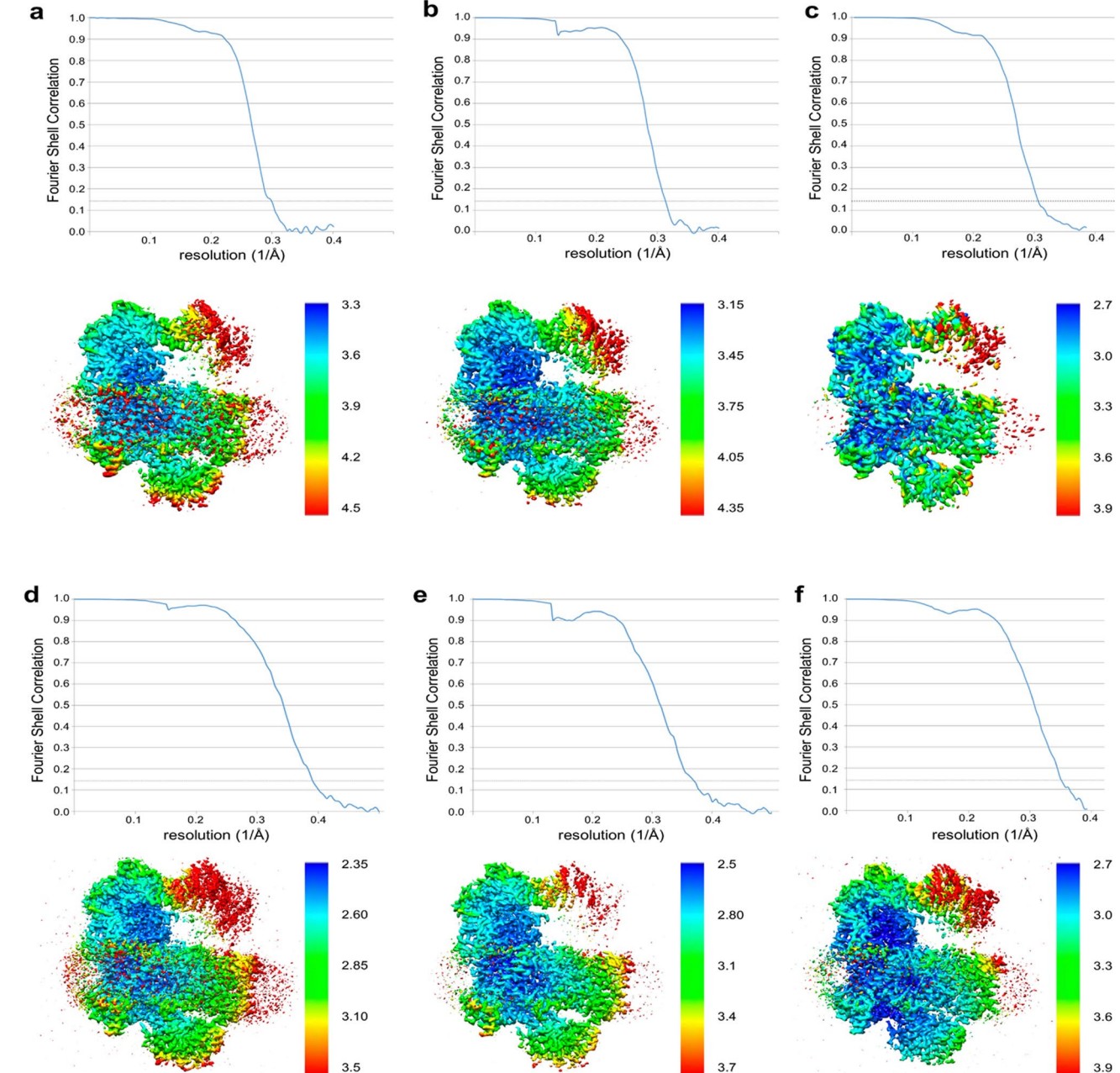

**Extended Data Fig. 2 | Cryo-EM data resolution.** Gold-standard Fourier Shell Correlation curves and maps colored by local resolution (color scale in Å). **a**, Native Na+-NQR, **b**, Na+-NQR with DQA, **c**, Na+-NQR with HQNO, **d**, Na+-NQR with UQ-1, **e**, Na+-NQR with UQ-2, **f**, Na+-NQR with NADH and UQ-2.

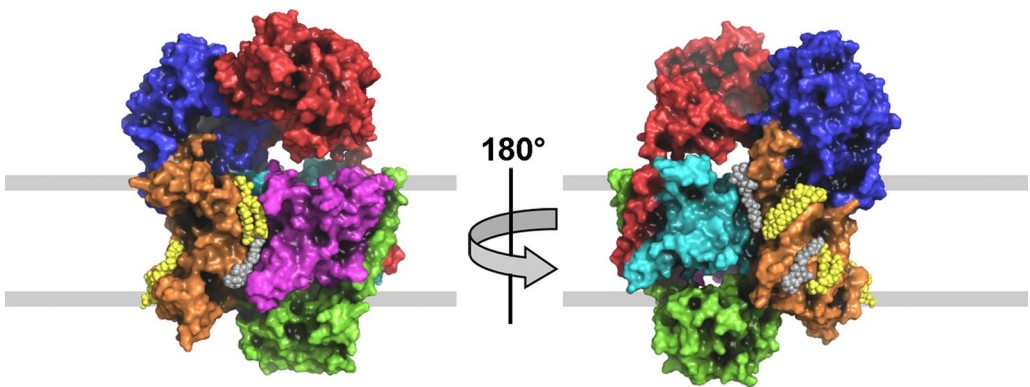

**Extended Data Fig. 3 | Location of lipids and detergent molecules bound to Na⁺-NQR.** Several lipids and DDM molecules were well defined by the cryo-EM density. The bound lipids and detergent molecules define the extent of membrane indicated here by grey lines. DDM detergent molecules are depicted in grey, lipid molecules in yellow. NqrA is shown in blue, NqrB in orange, NqrC in green, NqrD in magenta, NqrE in cyan and NqrF in red.

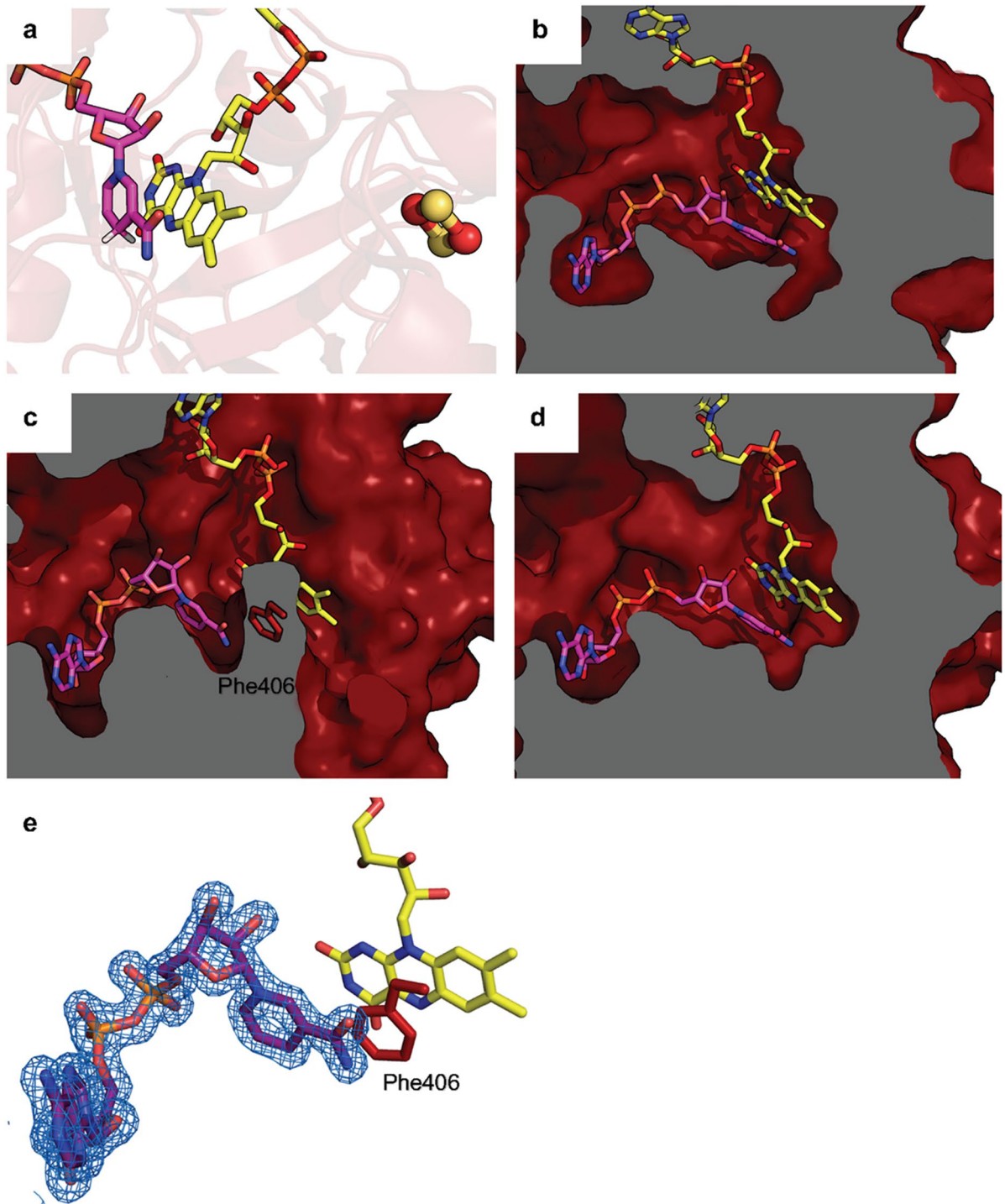

**Extended Data Fig. 4 | Structures of the FNR-like domain subunit NqrF in complex with NADH. a**, Complex of NADH with NqrF in Na$^+$-NQR cryo-EM structure. The C4N of the nicotinamide group of NADH carrying the 2 e$^-$ is at a distance of 3.3 Å of N5 of the FAD, that is in perfect hydride transfer distance. The edge-to-edge distance between the C8M of the FAD and the proximal Fe of the [2Fe$^-$2S] cluster is 9.4 Å. **b**, Binding pocket of NqrF in Na$^+$-NQR. The adenine moiety of NADH resides in a separate pocket of the protein, whereas the nicotinamide shares a pocket with the isoalloxazine of FAD. **c**, In contrast to the cryo-EM structure the X-ray structure of isolated NqrF in complex with NADH the nicotinamide is separated from the isoalloxazine by Phe406 and cannot transfer its hydride to N5 of FAD. Residue Phe406 has to move out of the binding pocket to promote hydride transfer. **d**, To address the role of residue Phe406 in NADH binding, we determined the X-ray structure of a NqrF variant where Phe406 is replaced by Ala. In contrast to wildtype NqrF the NqrF Phe406Ala variant exhibited productive binding of the nicotinamide moiety of NADH in close proximity to the FAD. The structure revealed that the nicotinamide binds close to the isoalloxazine. The C4N of the nicotinamide group of NADH is in a distance of 3.5 Å of N5 of the FAD. **e**, X-ray structure of NqrF subunit in complex with NADH shown in a similar orientation like in **c**. The 2*fo-fc* at 1.5 σ around bound NADH is shown.

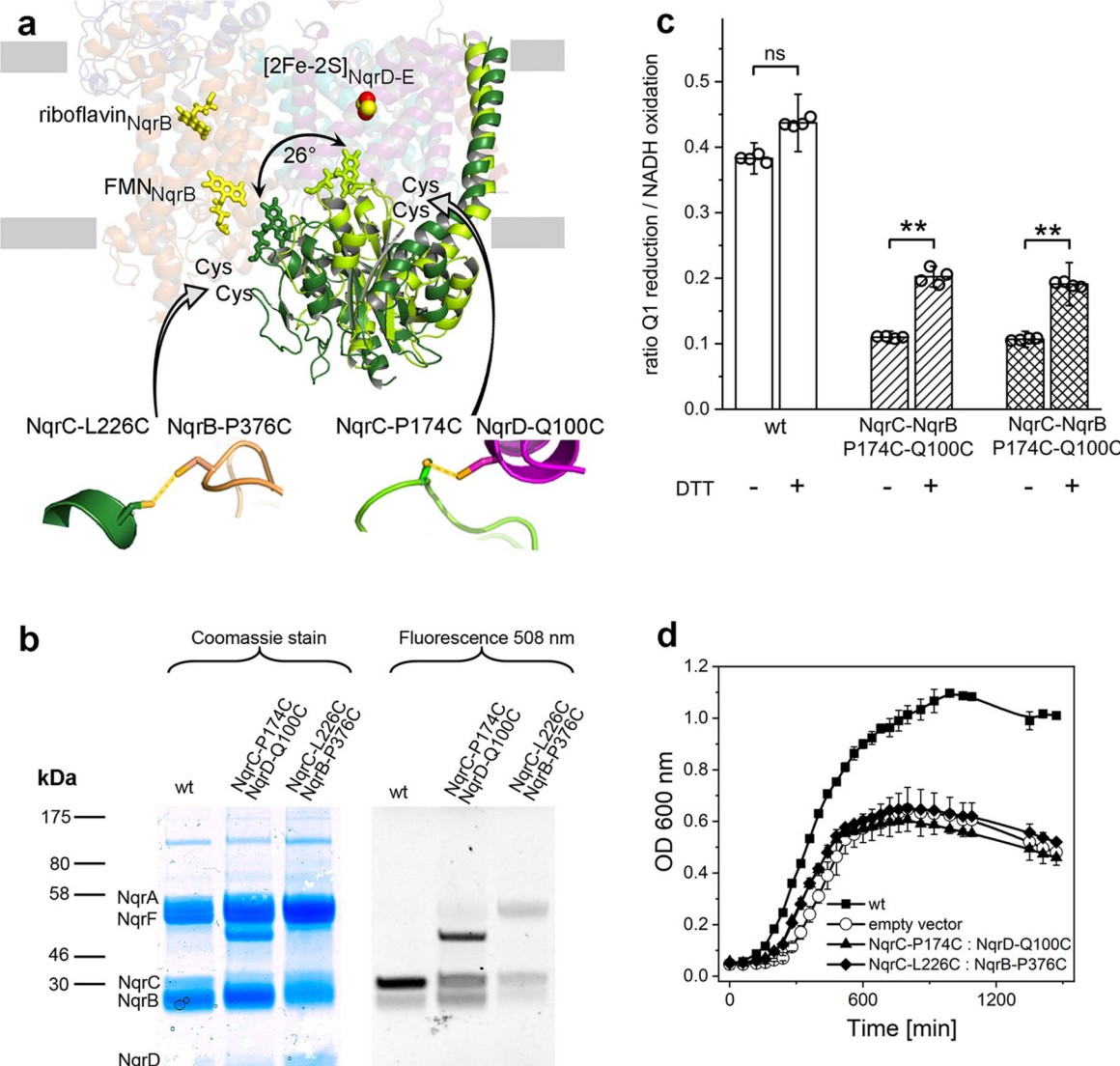

**Extended Data Fig. 5 | NqrC is locked by engineered disulfide bonds.**
**a**, The hydrophilic domain of NqrC moves between two positions at either
NqrB or NqrD-E shuttling the electrons from the intramembranous [2Fe⁻2S]
in NqrD-E to NqrB. The cryo-EM structures and the X-ray structures represent
snapshots of these switching movement of NqrC. On the basis of the structures,
cysteine residues were engineered in NqrC, NqrB and NqrD which are predicted
to form disulfide bonds in either state. The cysteine residues are located in
the periplasm that with its oxidizing environment facilitates disulphide bond
formation. **b**, Variants of Na⁺-NQR with corresponding cysteine residues were
isolated and analysed by non-reducing SDS-PAGE. Both variants corresponding
to both states of NqrC form disulphide bonds as indicated by the shift of bands
to higher molecular mass. The covalent FMN containing subunits NqrC and NqrB
were monitored by their fluorescence at 508 nm. The disulphide crosslink of
NqrC with NqrB and with NqrD demonstrates that both conformations occur

*in vivo*. **c**, Both variants showed diminished ratios of UQ-1/NADH reduction rates
showing that the electron transfer through Na⁺-NQR is impaired. Reduction
of the disulphide bonds restores the electron transfer to UQ-1, as shown by an
approximate 2-fold increase of the UQ-1/NADH ratios. Kinetic data are shown
as mean ± SD including error propagation, n = 4 independent experiments.
Individual data points are shown as open spheres. Two-sided paired sample *t* test
was applied with confidence interval = 0.95 and degree of freedom =6. The effect
sizes were 1.5 for wt, 7.2 for variant NqrC-P174C-NqrD-Q100C, and 3.4 for variant
NqrC-L226C-NqrB-P376C, respectively. The p-values are 0.076 for wt, $5.7110^{-5}$
for variant NqrC-P174C-NqrD-Q100C, and 0.003 for variant NqrC-L226C-NqrB-
P376C, respectively. **d**, *Vibrio cholerae* strains expressing the variants instead of
wt Na⁺-NQR are drastically impaired in growth displaying slower growth rates
and 45% lower cell yields. Growth data are shown as mean ± SD, n = 3 biologically
independent experiments.

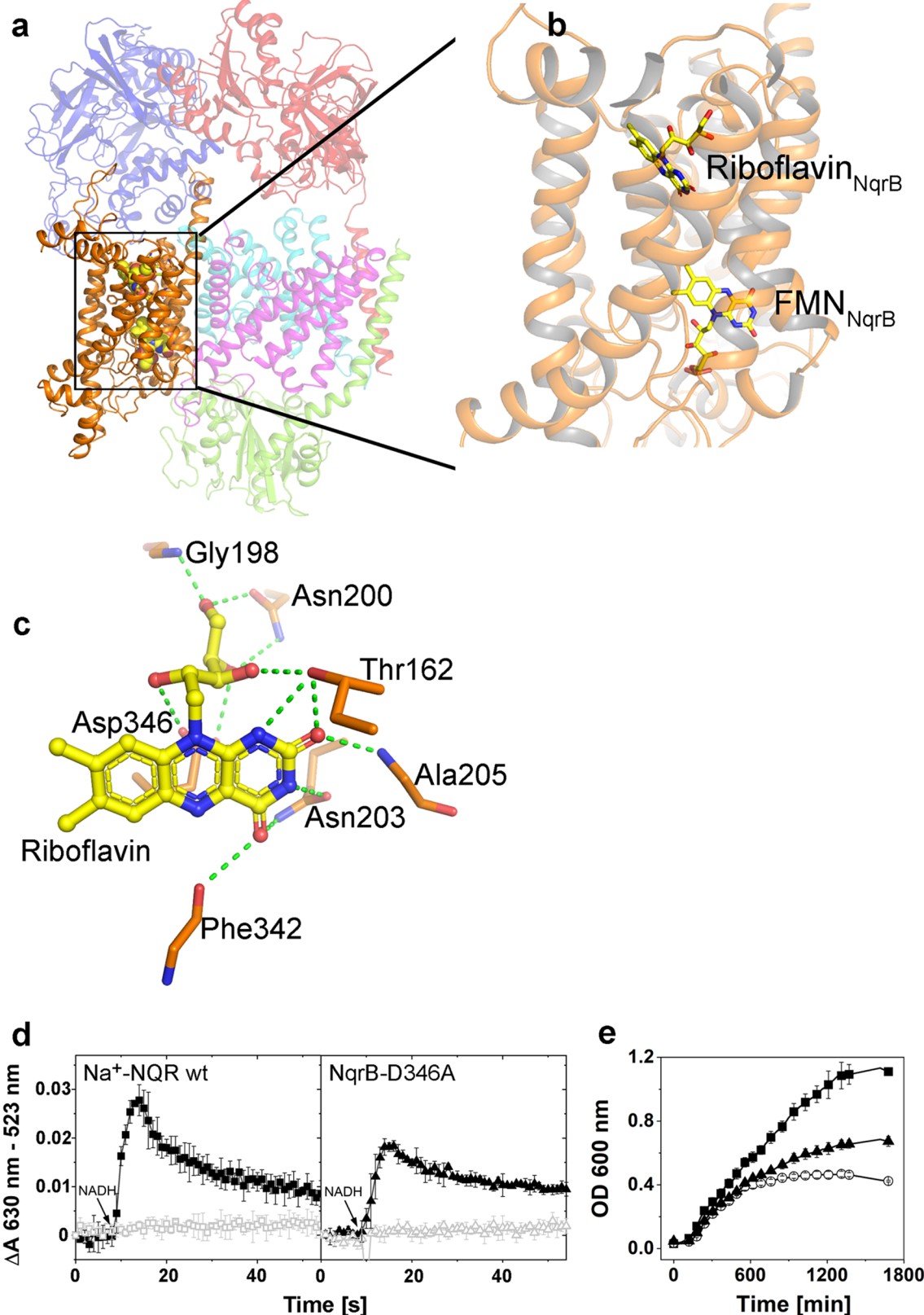

**Extended Data Fig. 6 | See next page for caption.**

**Extended Data Fig. 6 | Riboflavin site. a**, NqrB (orange) contains a covalent FMN and a riboflavin as cofactors. Both flavins are depicted as yellow spheres. **b**, Detailed view on the location of FMN and riboflavin in NqrB. The FMN cofactor is located close to the periplasmic aspect of NqrB, the riboflavin closer to the cytoplasmic aspect. **c**, Interactions of riboflavin with the residues of NqrB. Several hydrogen bonds (green dotted lines) are formed with the ribityl sidechain and the isoalloxazine. The backbone amide of Gly198 and the sidechains of Asn200, Thr162, and Asp346 form hydrogen bonds with the oxygen atoms of the ribityl chain. The backbone amide and carbonyl of Ala205 and Phe342 as well as

the sidechains of Thr162 and Asn203 form hydrogen bonds with N1, O2, N3 and O4 of the isoalloxazine. **d**, Voltage generation by isolated wt Na⁺-NQR and variant Na⁺-NQR-NqrB-D346A. Voltage generation by the variant is drastically decreased. Data are shown as mean ± SD, n = 3 independent experiments. **e**, Growth of *Vibrio cholerae* expressing either wt Na⁺-NQR (■), Na⁺-NQR-NqrB-346A variant (▲), or a Na⁺-NQR variant lacking the entire subunit NqrB (○). Growth rate and growth yield of Na⁺-NQR-NqrB-346A variant (▲) is diminished. Growth data are shown as mean ± SD, n = 3 biologically independent experiments.

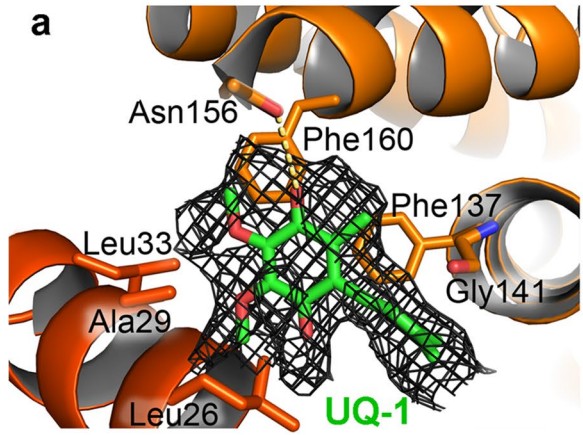

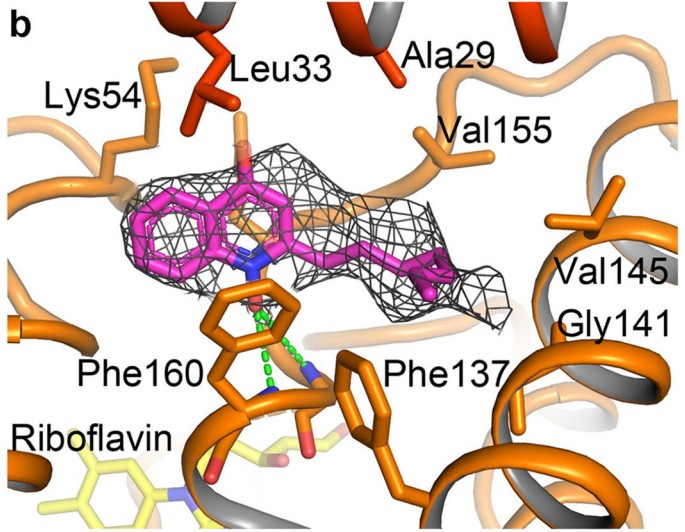

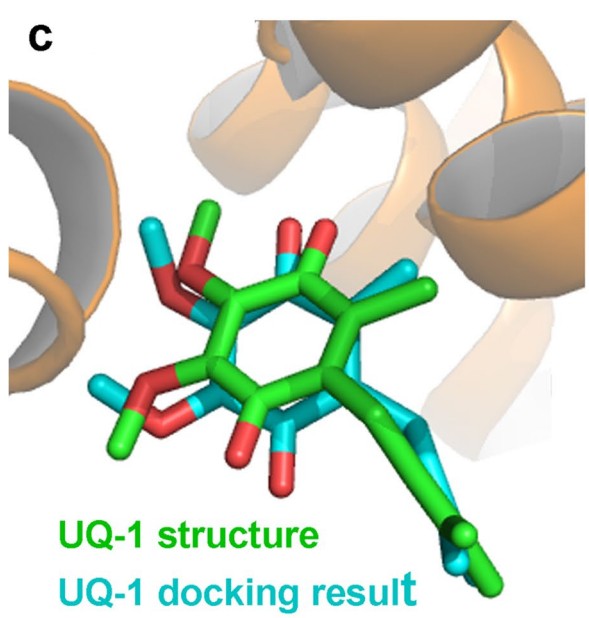

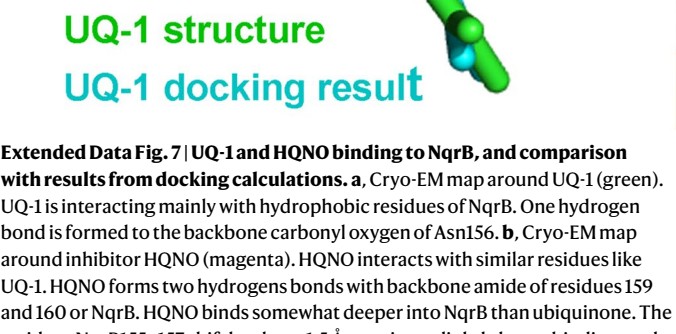

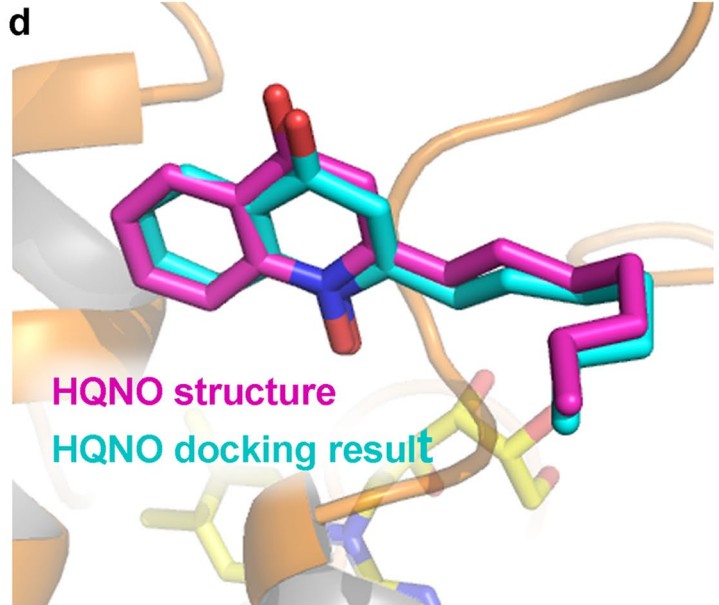

**Extended Data Fig. 7 | UQ-1 and HQNO binding to NqrB, and comparison with results from docking calculations. a**, Cryo-EM map around UQ-1 (green). UQ-1 is interacting mainly with hydrophobic residues of NqrB. One hydrogen bond is formed to the backbone carbonyl oxygen of Asn156. **b**, Cryo-EM map around inhibitor HQNO (magenta). HQNO interacts with similar residues like UQ-1. HQNO forms two hydrogens bonds with backbone amide of residues 159 and 160 or NqrB. HQNO binds somewhat deeper into NqrB than ubiquinone. The residues NqrB155–157 shift by about 1.5 Å creating a slightly larger binding pocket than the ubiquinone. Moreover, like the quinones, HQNO recruits the N-terminal amphiphilic helices to form a high affinity binding site. **c**, Docking of UQ1 into the structure of Na⁺-NQR yields a solution that is reasonable agreement with the modelled UQ-1. The docking energy is −5.6 kcal mol⁻¹. **d**, Docking of HQNO into the structure of Na⁺-NQR gives a perfect match of the docking solution and the modelled HQNO. The docking energy is −8.9 kcal mol⁻¹, that is HQNO is predicted to bind with higher affinity than UQ-1.

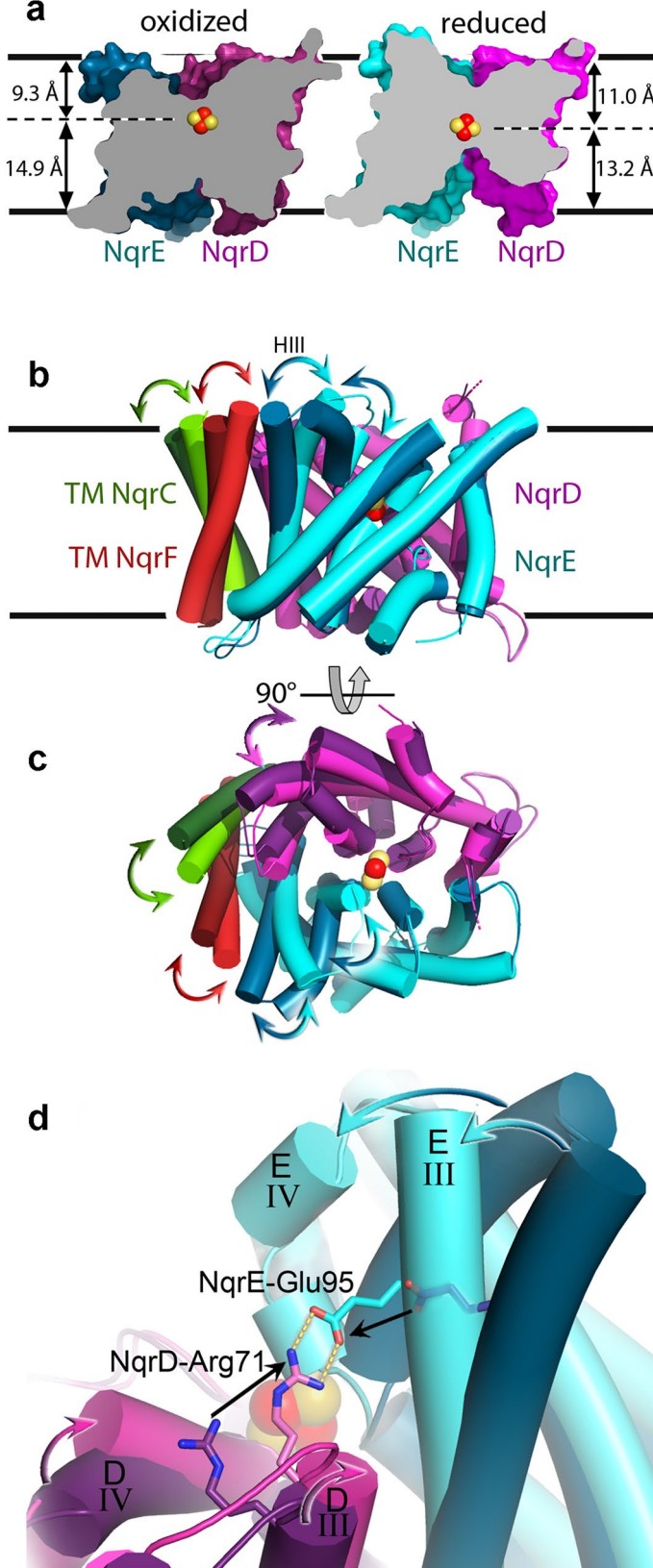

**Extended Data Fig. 8 | See next page for caption.**

**Extended Data Fig. 8 | Conformational changes in subunits NqrD and NqrE.**
**a**, Cross section through subunits NqrD-E at the position of the $[2Fe^-2S]_{NqrD\text{-}E}$ (shown as spheres). The cryo-EM structure of NqrD-E in the oxidized state is shown on the left-hand side in dark colors. The structure assigned to a reduced $[2Fe^-2S]_{NqrD\text{-}E}$ cluster is shown on the right-hand side in light colors. The distances of the cluster to the membrane planes are indicated. Switching from one state the other, the cluster moves almost 2 Å relative to the membrane plane. **b**, Structural alignment of NqrD-E and of the NqrC, NqrF transmembrane helices in both states. The conformational changes are indicated by arrows. **c**, as shown in **b** but in top view, illustrating how NqrD-E opens in the oxidized state (dark colors) compared to the reduced state (light colors). **d**, Detailed view of structural changes in the region of helices III and IV of NqrD-E. The movement of the helices between both states is indicated by arrows. NqrD-Arg71 of and NqrE-Glu95 approach each other to form a salt bridge stabilizing the conformation corresponding to the reduced state of the $[2Fe\text{-}2S]_{NqrD\text{-}E}$ cluster.

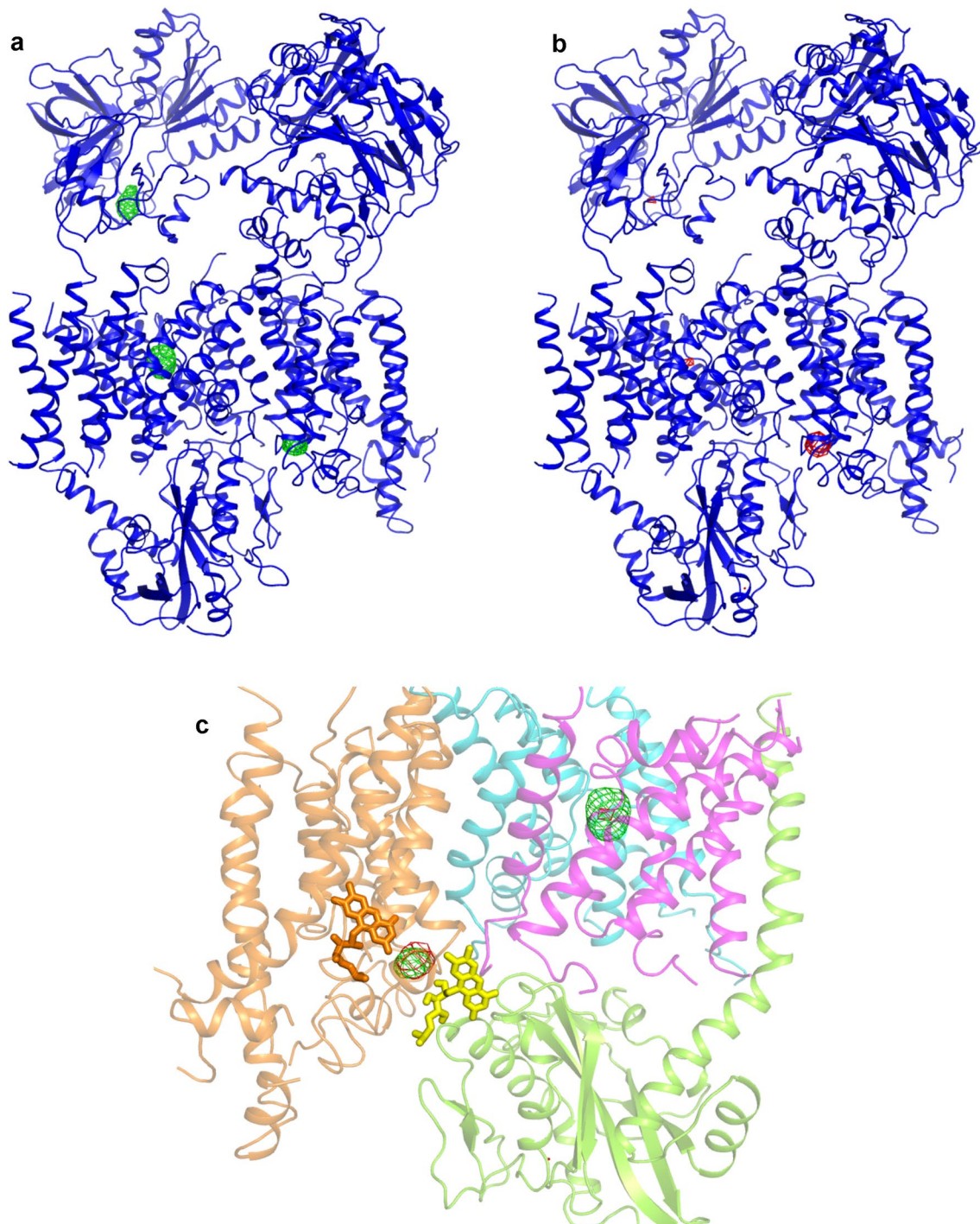

**Extended Data Fig. 9 | Localization of K⁺ site by Cs⁺ and Rb⁺. a**, Anomalous difference maps in the region of entire Na⁺-NQR. The map is derived from crystals grown in the presence of CsCl. The dataset was recorded at 1.7 Å close to the L-I edge of Cs to maximize the anomalous contribution of these metal ions. At this wavelength the [2Fe-2 S]$_{NqrF}$, the [2Fe-2S]$_{NqrD-E}$ and rigidly bound Cs⁺ will exhibit a strong anomalous signal. Strong anomalous difference map peaks (green) are observed at the positions of [2Fe-2S]$_{NqrF}$, [2Fe-2S]$_{NqrD-E}$ and at the periplasmic aspect of NqrB. **b**, Anomalous difference maps of Na⁺-NQR derived from crystals grown in the presence of RbCl. The dataset was recorded at the K-edge of Rb at 0.82 Å. Strong anomalous difference map peaks (red) are observed at the periplasmic aspect of NqrB. Some minor peaks deriving from the [2Fe-2S] clusters are observed as well. **c**, Overlay of the anomalous difference maps deriving from Cs⁺ (green) and Rb⁺ (red) bound to Na⁺-NQR. The binding site for both ions maps to site Na-2 close to the interaction site of NqrB with NqrC.

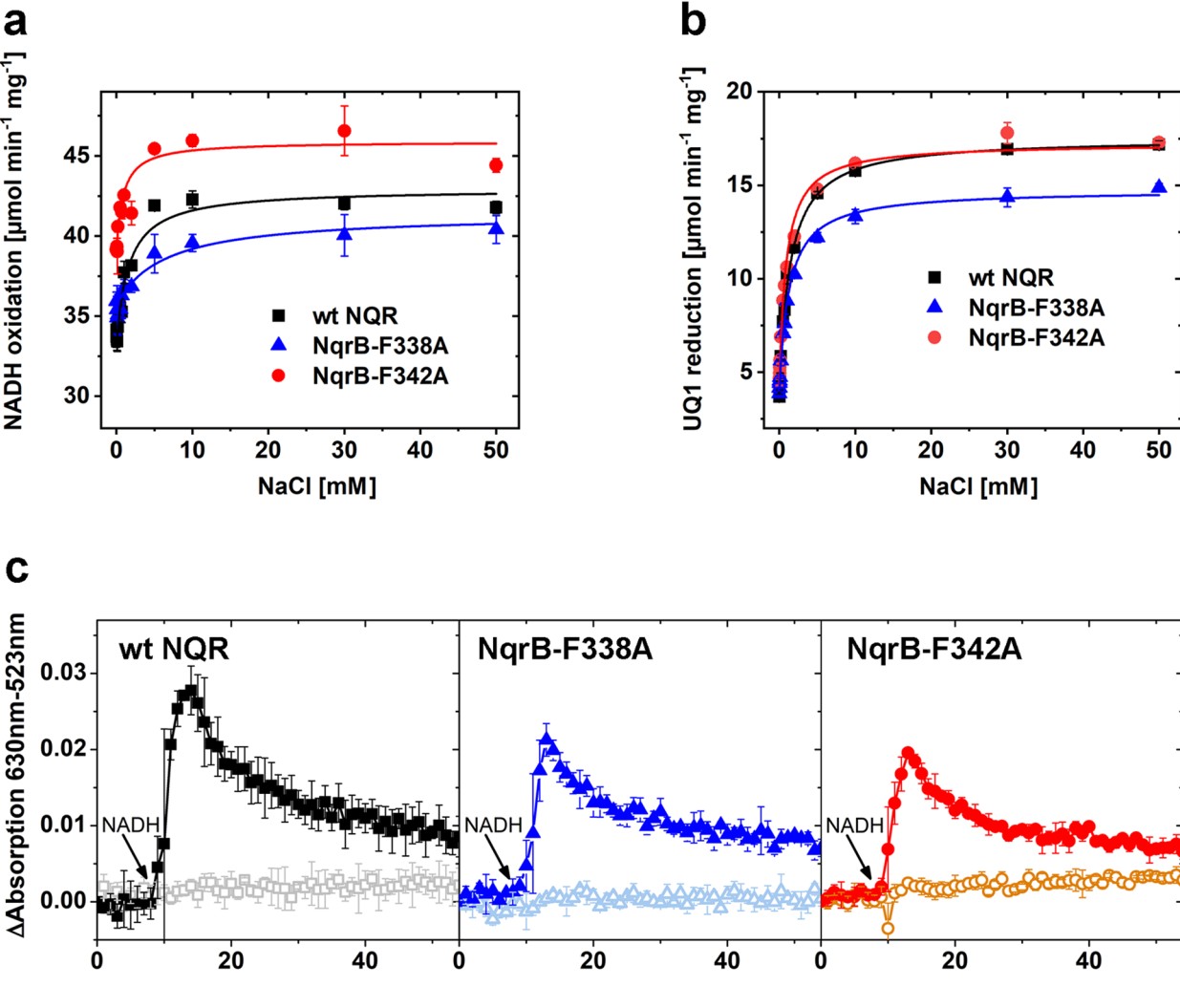

**Extended Data Fig. 10 | Kinetic data and voltage generation by wildtype Na⁺-NQR and Na⁺-NQR variants. a**, Na⁺-dependent NADH oxidation rates of wt Na⁺-NQR and variants NqrB-F338A and NqrB-F342A. The variant exhibits an NADH oxidation rate that is about 5% diminished compared to wt Na⁺-NQR, whereas variant NqrB-F342A shows a 7% higher activity than wt NQR. **b**, Na⁺-dependent UQ1 reduction rates reveal almost identical rates for wt Na⁺-NQR and variant NqrB-F342A, while the rate for NqrB-F338A is reduced by about 8%. In summary, electron transfer activity is only marginally affected in both variants confirming that the structural integrity is not affected by these mutations. Data are shown as mean ± SD, n = 3 independent experiments.**c**, Voltage generation in proteoliposomes monitored with oxonol is reduced by about 30% for variants NqrB-F338A and NqrB-F342A compared to wt Na⁺-NQR. Residues Phe338 and Phe342 are located at the constriction of the proposed Na⁺-translocation pathway and might serve as a gate. In both variants the bulky Phe residue has been replaced by the smaller Ala, respectively, resulting in decreased voltage generation. This may be explained by a backflow of Na⁺ during translocation because of an incomplete closure of the gate. In control experiments without addition of NADH (dotted trace), no voltage formation is observed. Data are shown as mean ± SD, n = 3 independent experiments.

# Reporting Summary

## Statistics

For all statistical analyses, confirm that the following items are present in the figure legend, table legend, main text, or Methods section.

| n/a | Confirmed | |
|---|---|---|
| ☐ | ☒ | The exact sample size (*n*) for each experimental group/condition, given as a discrete number and unit of measurement |
| ☐ | ☒ | A statement on whether measurements were taken from distinct samples or whether the same sample was measured repeatedly |
| ☐ | ☒ | The statistical test(s) used AND whether they are one- or two-sided<br>*Only common tests should be described solely by name; describe more complex techniques in the Methods section.* |
| ☒ | ☐ | A description of all covariates tested |
| ☒ | ☐ | A description of any assumptions or corrections, such as tests of normality and adjustment for multiple comparisons |
| ☐ | ☒ | A full description of the statistical parameters including central tendency (e.g. means) or other basic estimates (e.g. regression coefficient) AND variation (e.g. standard deviation) or associated estimates of uncertainty (e.g. confidence intervals) |
| ☐ | ☒ | For null hypothesis testing, the test statistic (e.g. *F*, *t*, *r*) with confidence intervals, effect sizes, degrees of freedom and *P* value noted<br>*Give P values as exact values whenever suitable.* |
| ☒ | ☐ | For Bayesian analysis, information on the choice of priors and Markov chain Monte Carlo settings |
| ☒ | ☐ | For hierarchical and complex designs, identification of the appropriate level for tests and full reporting of outcomes |
| ☒ | ☐ | Estimates of effect sizes (e.g. Cohen's *d*, Pearson's *r*), indicating how they were calculated |

*Our web collection on statistics for biologists contains articles on many of the points above.*

## Software and code

Policy information about availability of computer code

| Data collection | Data were collected at synchrotron sources: 1) PETRA-3 beamline P14 using MXCUBE  2) at Swiss-Light -Source beamlines X06SA and X06DA using DA+ for data acquisition. Cryo-EM images were recorded using EPU (version 2.1). |
|---|---|
| Data analysis | All X-ray images were processed with XDS (version Jan 10, 2022  BUILT=20220820) and scaled with XSCALE (version Jan 10, 2022 BUILT=20220820) . For analysis of anomalous scattering the program anode was used. Molecular replacement was performed with Phaser (version 2.8.3). X-ray structures were refined using Refmac5 (version 5.8.350) and phenix.refine (version 1.20.1) and Coot (version 0.9.8.1) was used for manual rebuilding. Anomalous difference maps were calculated with the program anode (version 2013/1).<br>Cryo-EM images were processed using RELION4. Particles were picked using Topaz (version 0.2.4).Contrast transfer function was estimated using CTFFIND4.1 inside RELION4. Cryo-EM maps were sharpened with LocScale and phenix.resolve_cryo_em (version 1.20.1). Partial models were generated using Alphafold2. The models were refined using phenix.real_space_refine (version 1.20.1) and Coot (version 0.9.8.1)  for manual rebuilding. Restraints used during refinement for ubiquinone-1, ubiquinone-2, HQNO, and riboflavin were generated with acedrg (version 246) from smiles codes; for energy minimization Refmac5 (version 5.8.0350) was used. Restraints for the covalent link between FMN and threonine were generated using acedrg. The cif file was edited manually using angles and distances obtained after energy minimization with gamess (version 18 AUG 2016 (R1)). Harmonic restraints used during refinement in phenix.refine (version 1.20.1) and phenix.real_pace_refine7 (version 1.20.1) for the coordination and geometry of the [2Fe 2S] clusters and coordination of the Na+ ions were generated by phenix.elbow (version 1.20.1) Putative ions in the structures were analyzed with WASP (version 1.0). Putative ion channels were identified with HOLE (version 2.2.005). Docking calculations were performed with PLANTS (version 1.2), SMINA (version Oct 15 2019,  based on AutoDock Vina 1.1.2) and VINAXB (based on AutoDock Vina 1.1.2) . Cross-linked peptides from mass spec analysis were identified using xQuest (version 2.1.5) and quantification of cross-links was performed using xTract (version 1.0.2).  KM values were fitted using Origin 2019. |

For manuscripts utilizing custom algorithms or software that are central to the research but not yet described in published literature, software must be made available to editors and reviewers. We strongly encourage code deposition in a community repository (e.g. GitHub). See the Nature Portfolio guidelines for submitting code & software for further information.

## Data

Policy information about availability of data

All manuscripts must include a data availability statement. This statement should provide the following information, where applicable:
- Accession codes, unique identifiers, or web links for publicly available datasets
- A description of any restrictions on data availability
- For clinical datasets or third party data, please ensure that the statement adheres to our policy

Cryo-EM density maps have been deposited in the Electron Microscopy Data Bank under accession numbers EMD-15088 (Na+-NQR native), EMD-15091 (Na+-NQR with ubiquinone-1), EMD-15090 (Na+-NQR with ubiquinone-2), EMD-15089 (Na+-NQR with ubiquinone-2 and NADH), EMD-15092 (Na+-NQR with DQA), EMD-15093 (Na+-NQR with HQNO). Cryo EM model coordinates have been deposited in the Protein Data Bank under accession numbers 8A1T (Na+-NQR native), 8A1W (Na+-NQR with ubiquinone-1), 8A1V (Na+-NQR with ubiquinone-2), 8A1U (Na+-NQR with ubiquinone-2 and NADH), 8A1X (Na+-NQR with DQA), 8A1Y (Na+-NQR with HQNO).
X-ray structure coordinates and structure factors for the entire complex of Na+-NQR and of individual subunit NqrF (residues 129–408) and NqrF-F406A variant (residues 129–408) with and without substrate NADH have been deposited in the Protein Data Bank. The PDB accession codes are 8ACY (Na+-NQR updated entry), 8ACW (Na+-NQR new entry), 8ACY (Na+-NQR with DQA), 8AD4 (subunit NqrF with NADH), 8AD3 (subunit NqrF-F406A), 8AD5 (subunit NqrF-F406A with NADH).
The cross-linking mass spectrometry data have been deposited to the ProteomeXchange Consortium via the PRIDE48 partner repository with the dataset identifier PXD039289.
The structure of Na+-NQR (pdb code 4P6V) was used as an initial model for model building of cryo-EM and X-ray structures.  For molecular replacement the structure of NqrF FAD domain (pdb code 4U9U) was used as  search model. Models were retrieved from the pdb database (https://www.rcsb.org/)

## Research involving human participants, their data, or biological material

Policy information about studies with human participants or human data. See also policy information about sex, gender (identity/presentation), and sexual orientation and race, ethnicity and racism.

| | |
|---|---|
| Reporting on sex and gender | does not apply |
| Reporting on race, ethnicity, or other socially relevant groupings | does not apply |
| Population characteristics | does not apply |
| Recruitment | does not apply |
| Ethics oversight | does not apply |

Note that full information on the approval of the study protocol must also be provided in the manuscript.

# Field-specific reporting

Please select the one below that is the best fit for your research. If you are not sure, read the appropriate sections before making your selection.

☒ Life sciences        ☐ Behavioural & social sciences        ☐ Ecological, evolutionary & environmental sciences

For a reference copy of the document with all sections, see nature.com/documents/nr-reporting-summary-flat.pdf

# Life sciences study design

All studies must disclose on these points even when the disclosure is negative.

| | |
|---|---|
| Sample size | Sample size is defined by the samples themselves. Crystals for X-ray analysis contain billions of molecules. Cryo-EM single particle requires several hundred thousands molecules. |
| Data exclusions | No data were excluded. In X-ray data integration severe outliers of spots intensity are identified by data processing software and excluded. These are usually only in the range of tens to few hundreds diffraction spots compared to millions of spots integrated per dataset. |
| Replication | Experimental procedures are highly reproducible. X-ray structure analysis involved measurement of several hundred crystals with highly similar properties. Cryo-EM analysis was performed with several grids, each containing several hundreds to thousands holes. |
| Randomization | Randomization was not relevant for this study. Samples are highly homogeneous. |

| Blinding | Blinding was not relevant for this study. Samples are highly homogeneous. |

# Reporting for specific materials, systems and methods

We require information from authors about some types of materials, experimental systems and methods used in many studies. Here, indicate whether each material, system or method listed is relevant to your study. If you are not sure if a list item applies to your research, read the appropriate section before selecting a response.

## Materials & experimental systems

| n/a | Involved in the study |
|-----|----------------------|
| ☒ ☐ | Antibodies |
| ☒ ☐ | Eukaryotic cell lines |
| ☒ ☐ | Palaeontology and archaeology |
| ☒ ☐ | Animals and other organisms |
| ☒ ☐ | Clinical data |
| ☒ ☐ | Dual use research of concern |
| ☒ ☐ | Plants |

## Methods

| n/a | Involved in the study |
|-----|----------------------|
| ☒ ☐ | ChIP-seq |
| ☒ ☐ | Flow cytometry |
| ☒ ☐ | MRI-based neuroimaging |

