## [Peer Review File · Nature Structural & Molecular Biology]

Peer Review Information

Manuscript Title: Conformational coupling of redox-driven Na⁺-translocation in *Vibrio cholerae* NADH:quinone oxidoreductase

Corresponding author name(s): Janet Vonck, Julia Steuber, Guenter Fritz

Reviewer Comments & Decisions:

Decision Letter, initial version:
--

Message: 7th Oct 2022

Dear Dr. Fritz,

Thank you again for submitting your manuscript "Conformational coupling of redox-driven Na⁺-translocation in *Vibrio cholerae* NADH:quinone oxidoreductase". I apologize for the delay in responding, which resulted from the difficulty in obtaining suitable referee reports. Nevertheless, we now have comments (below) from the 3 reviewers who evaluated your paper. In light of those reports, we remain interested in your study and would like to see your response to the comments of the referees, in the form of a revised manuscript.

Please be sure to address/respond to all concerns of the referees in full in a point-by-point response and highlight all changes in the revised manuscript text file. If you have comments that are intended for editors only, please include those in a separate cover letter.

We expect to see your revised manuscript within 6 weeks. If you cannot send it within this time, please contact us to discuss an extension; we would still consider your revision, provided that no similar work has been accepted for publication at NSMB or published elsewhere.

Reporting Summary:

Please note that all key data shown in the main figures as cropped gels or blots should be presented in uncropped form, with molecular weight markers. These data can be aggregated into a single supplementary figure item. While these data can be displayed in a relatively informal style, they must refer back to the relevant figures. These data should be submitted with the final revision, as source data, prior to acceptance, but you may want to start putting it together at this point.

Data availability: this journal strongly supports public availability of data. All data used in accepted papers should be available via a public data repository, or alternatively, as Supplementary Information. If data can only be shared on request, please explain why in your Data Availability Statement, and also in the correspondence with your editor. Please

note that for some data types, deposition in a public repository is mandatory - more information on our data deposition policies and available repositories can be found below: <https://www.nature.com/nature-research/editorial-policies/reporting-standards#availability-of-data>

[Redacted]

Sincerely,

Carolina Perdigoto, PhD
Chief Editor
Nature Structural & Molecular Biology
orcid.org/0000-0002-5783-7106

Reviewers' Comments:

Reviewer #1:

Remarks to the Author:

The manuscript summarises large amount of structural and functional data on sodium-translocating NADH:quinone oxidoreductase, allowing authors to elaborate the mechanism of coupling between electron transfer reactions and Na⁺ translocation. This is a major achievement in bioenergetics field. In its details the mechanism still remains somewhat speculative, especially on how exactly Na⁺ translocation is achieved, but solid foundations for future work on these details are provided here. Overall the manuscript is well written and illustrated, and will be of wide interest to NSMB readers.

Specific points to take into account in revision:

P. 4 – Conformational changes in NqrF subunit are discussed but lack detail. This important information should be illustrated with (overlaid) structures as one of the main figures, not just in Supplement. Importantly, it should be stated under which conditions a particular conformation is observed – currently this key information is not presented here, only much later in discussion.

P. 6, Fig. 2 e,f – it is not clear if these panels show NQR only with NqrD-E cluster or with both clusters (also NqrF)? Fig. 2f – how was the cluster reduced? Its spectrum looks very different from usual 2Fe-2S clusters, i.e. no clear g-xyz pattern. Why? How does NqrF cluster spectrum looks like? This should be illustrated and discussed in detail.

P. 7 – similarly to NqrF above, it should be indicated in which experimental conditions a particular NqrC conformation was observed.

P. 6, P.8, Suppl Data Table 2 – it is not clear why the rate of UQ1 reduction is only about 30% or so of the rate of NADH oxidation even in WT? It should be close to 100%. Where do the rest of electrons from NADH go (or leak) to? This should be discussed.

ED Fig. 5d and 9 – the legends should state how voltage was measured here (oxonol).

Fig. 4 – should show also how UQ2 binds, since structures were determined.

P. 11, l. 2 – why NADH and UQ2 together were used to create reduced state? Just NADH should be enough. Addition of NADH together with UQ2 will create turnover-like conditions – this should be discussed in detail. It appears that one of the key states (fully reduced by NADH) may be missing from the current structural analysis. The relocation of NqrF cluster towards NqrD-E might be then observed? To refer to the current state in l. 21 as a “reduced state” seems to be wrong, it probably applies only to NqrD-E cluster.

P. 11, l. 22 – experimental structures of oxidised and “reduced”-like state do not differ in NqrC position, so this comment is not clear.

Reviewer #2:

Remarks to the Author:

Hau et al present a series of cryo-EM and X-ray structures of intermediates of the Na⁺-pumping Na⁺-NQR from *Vibrio cholerae* together with a quite large array of data from complementary techniques to verify/support their findings.

The data are of good quality and the topic interesting and novel insights should be possible to extract. However, the manuscript as written needs to be significantly revised, especially in terms of clarifying the resulting model of redox-driven Na⁺-translocation. Specific comments in order of when they appear in the manuscript is found below:

1. Introduction to 'state of the art' is very short and gives very little background to what was known and where current work fits in.
2. Also it is hard to understand how much is new in this study compared to the previous X-ray structure, which should be more clearly described.
3. On page three the previous X-ray structure at 3.5 Å is said to be of low resolution compared to the now obtained cryo-EM structures at 2.1-3.4Å. This seems to be a statement that needs modification and explanation.
4. For the structure of Na⁺-NQR in complex with NADH, it is unclear why there has been no electron transfer from NADH?
5. On p.7 The discussion on ET distances is a bit oversimplified. What would be the rate of ET at 22Å and how much slower is that compared to turnover of the enzyme?
6. The large conformational changes that are observed needs to go in a clear model for what their role would be and this model should be presented in the proper manuscript, see below.
7. UQ-1 and UQ-2 were localized, but what is the native quinol used in the *Vibrio* cells?(goes back to the lack of introduction)
8. On p.9 the shallow quinol binding site compared to complex I is discussed, but in the case of complex I this feature has been suggested to be involved in the pumping mechanism, so how is this related to the mechanism envisaged here?
9. The description of non-competitive inhibition of HQNO is confusing, is it competitive or not? If HQNO exhibits truly non-competitive behaviour, they should be able to bind both at the same time, true here?
10. On p.10 it is stated that the authors 'develop a model of redox-driven Na⁺ translocation' which is very good. However, the model itself is not presented as a figure in the main manuscript which it absolutely be.
11. On p. 11, it's a bit unclear why the different states observed are assigned to the redox states they are (in general) and why this could not be verified by spectroscopy as the additions are made well in advance of freezing the samples for cryo-EM? Further down, one state is referred to as the 'reduced' state in quotation marks. Why could the state obtained not be verified?
12. The order of results does not make sense. There should be a description of what Na sites etc are seen first and what kind of changes are seen in the different states described before (+/- NADH etc) before trying to compile into a catalytic cycle, and the cycle needs to contain a model for how the changes observed translate into Na⁺ pumping, if the title of 'redox-driven Na⁺ translocation' should be relevant.
13. On p.14 there is no description of the Na⁺ pathway, only the binding site. What lines the pathway? Is it conserved?
14. The usage of site-directed variants is not properly described. What is the catalytic turnover in the Phe38 and Phe342 variants? The smaller voltage generation is no way is proof enough that effects seen are due to impairment of Na⁺ translocation per se. It needs to be compared to the rate of NADH oxidation and controlled for 'active' enzyme as this can vary extensively when making mutations. Also the protocol in general for making

vesicles seems rather 'harsh' and there should be statements of catalytic rates before and after to verify that the enzyme stays as active after the treatment.

15. On page 14-15 the general workings of a redox-driven ion pump is way too much over-simplified. In no way does such a system need to 'set pauses' between electron transfer steps in the way described. The system of course needs to couple ET to uptake/release/transfer of the ion but this can be achieved in many ways, e.g. by controlling the redox potentials such that the ion needs to be moved before ET can occur. This can be done without changing the distance between redox cofactors at all, as exemplified for instance by the cytochrome oxidase proton pump. The section needs to be heavily revised.

16. In the resulting model (p. 15) which should be outlined in the main text, it should also be made more clear which statements have support in the data and which do not.

17. On line 18 p15, 'opening of the gate' refers to which gate?

Reviewer #3:

Remarks to the Author:

The human pathogen *Vibrio cholerae* relies on the Na⁺-translocating NADH:quinone oxidoreductase (Na⁺-NQR) to generate membrane potential for energy metabolism. It is estimated that Na⁺-NQR accounts for over 80% of Na⁺ pumping activity and voltage formation, making it an attractive target for antibiotic development. The Gunter Fritz lab published a crystallographic study of the *Vibrio cholera* complex in 2014. This manuscript by Hau et al. expands on and improves the previous study of the same complex. The series of structures of the Na-NQR determined in different ligand binding states highlight conformational changes that are presumably linked to the enzyme functional cycle. They complemented their impressive structural studies with a comprehensive biophysical characterization, demonstrating an intimate coupling of protein conformational changes with sodium translocation and electron transfer process. The authors further augmented their findings with mutagenesis and cross-linking experiments. The structural work is well done, and the mechanistic proposal is interesting and supported by the evidence presented.

Major point

The authors combined spectroscopy and high-resolution structural data to show an imbedded [2Fe-2S] co-factor in the membrane region. However, the presence of such co-factor has been reported in a recent paper by Kishikawa et al. in *Nature Communication* (2022). The Kishikawa paper also reported conformations similar to those in the current manuscript. They need to cite the literature and compare and discuss when appropriate.

Minor points:

1. Please consider presenting cryo-EM data processing flowchart.
2. Because density modification made a difference, it would be nice to show a side-by side comparison of before and after densities in key regions.
3. Fig. 1d, the distance between the riboflavin and FMNNqrB is labeled incorrectly. It should be 29.3 Å according to the previous literature.
4. Fig. 3c, the two mutants have the same label. One of them should be L226C/P376C.
5. Please provide more details in figure legends.
6. Figure 2c: MCD data are usually presented with y-axis in unit of M⁻¹ cm⁻¹ T⁻¹, please also specify the magnetic field strength in the methods section.
7. The growth curve data shown in Fig. 2g was not discussed in the main text.

8. Page 2, Line 6: the word should be "unprecedented", not "unprecedent".
9. Page 4 line 24 - "extrapolatation" - check spell.

Reviewer #4:
None

Author Rebuttal to Initial comments

Point-by-point answer to reviewers' comments of manuscript NSMB-A46673-T

We thank all reviewers for their positive evaluation and detailed comments which helped us to improve the manuscript.

Reviewer #1:

The manuscript summarises large amount of structural and functional data on sodium-translocating NADH:quinone oxidoreductase, allowing authors to elaborate the mechanism of coupling between electron transfer reactions and Na⁺ translocation. This is a major achievement in bioenergetics field. In its details the mechanism still remains somewhat speculative, especially on how exactly Na⁺ translocation is achieved, but solid foundations for future work on these details are provided here. Overall the manuscript is well written and illustrated, and will be of wide interest to NSMB readers.

We thank Reviewer #1 for acknowledging the amount of data we are providing here.

Specific points to take into account in revision:

P. 4 – Conformational changes in NqrF subunit are discussed but lack detail. This important information should be illustrated with (overlaid) structures as one of the main figures, not just in Supplement. Importantly, it should be stated under which conditions a particular conformation is observed – currently this key information is not presented here, only much later in discussion.

We have included a panel in Figure 1 now illustrating the major changes in entire subunit NqrF (Fig 1c). Please note we just show two structures aligned. If we would align more structures the changes would be very difficult to recognize. Moreover, we have modified the Figure in Supplementary data and included now a Figure showing the conformation of the ferredoxin-like domain relative to the transmembrane helix in the different cryo EM and X-ray structures determined.

Thank you for pointing us to the unclear point on the different conditions. This is now clearly stated in the main text on page 5 and top of page 6 as well as in the figure legend of Fig. 1 of the revised manuscript.

Modified figures:

Figure 1c, structural overlay of subunit NqrF from cryo-EM structure with NADH and Q2 (red) and X-ray structure (grey). Subunit NqrD (magenta) and NqrE (cyan) are shown as surface. In comparison to the X-ray structure, the FNR-like domain of NqrF is shifted sideways and the ferredoxin-like domain is rotated towards the membrane subunits in the cryo-EM structure.

Supplementary Data Figure 2. Structural snapshots of rotational flexibility of the ferredoxin-like domain of NqrF. The ferredoxin-like domain of NqrF resides at different positions in the different cryo-EM and X-ray structures. **a**, Left: Structural alignment of the NqrF ferredoxin domain and the transmembrane helix domain. The FAD containing FNR-like domain is omitted for clarity. There is a rotational and a transversal movement of the ferredoxin domain relative to the transmembrane helix. For clarity, only the X-ray structure (orange), the cryo-EM structure with inhibitor HQNO (red) and the X-ray structure with inhibitor DQA (dark red) are shown. Right: extrapolation (grey) of the movement of the ferredoxin-like domain towards the membrane plane. **b**, Different conformations of the NqrF ferredoxin domain observed in this study with respect to the transmembrane helix. Broken lines indicate most distant and most proximate position of ferredoxin domain from the membrane, and most distant and proximate position of the [2Fe-2S] cluster from the membrane.

P. 6, Fig. 2 e,f – it is not clear if these panels show NQR only with NqrD-E cluster or with both clusters (also NqrF)? Fig. 2f – how was the cluster reduced? Its spectrum looks very different from usual 2Fe-2S clusters, i.e. no clear g -xyz pattern. Why? How does NqrF cluster spectrum look like? This should be illustrated and discussed in detail.

The panels 2 e,f show only the features of [2Fe-2S] cluster in NqrD-E. A variant lacking the cluster in NqrF was used to study the properties of [2Fe-2S] cluster in NqrD-E only. We have modified the figure legend to state this more clearly. To add clarity to the figure, as suggested by the reviewer, panels e and f are now labelled with the cluster name.

We have modified the text to include additional details and description of the EPR experiment. We appreciate the reviewer's comments about the unusual EPR pattern. Unfortunately, we had included an incorrect plot. The initial plot was of the raw, as-collected EPR spectrum of the sample. The weak signal is somewhat distorted by a copper-like background signal that is observed at low temperatures and high microwave powers, as employed here. This background arises from the cavity itself and is observed in a background scan of a blank sample of water taken under the same instrumental conditions. Subtraction of the background yields the now presented EPR spectrum. We have included the raw sample and background here in our reviewer responses. We apologize for the confusion.

We have added references to the well-characterized NqrF cluster spectrum that was previously published. Additionally, we have now included a simulation of the EPR spectrum recorded to make the cluster's spectrum more clear and distinguish it from the minor radical signal.

Modified text:

“EPR spectroscopy of the dithionite reduced intramembranous cluster in NqrD-E exhibited a weak signal with two distinct features. The prominent feature at $g \sim 2.01$ is consistent with a microwave power saturated radical signal (Fig. 2f), while the feature at $g \sim 1.94$ was only observed under relatively high microwave powers, indicative for a fast-relaxing species. The high-field feature resembles the g_{\perp} of various [2Fe-2S] clusters, however, a distinct corresponding g_{\parallel} feature of an axial EPR spectrum in the expected range of 2.01 to 2.06 is not observed. The EPR spectrum is adequately reproduced by simulation with two components: an isotropic radical at $g \sim 2.01$ and a dominant axial component with $g_{\parallel} = 2.02$ and $g_{\perp} = 1.94$ (Fig. 2f). The assigned g -values of the NqrD-E cluster are very similar to that observed for the NqrF cluster, but we do note the potentially very different relaxation behaviours of the two clusters as indicated by differences in microwave power required for observation.”

New figures:

Figure 2f. 10 K X-band EPR spectrum of Na^+ -NQR with $[\text{2Fe-2S}]_{\text{NqrD-E}}$ cluster simulated with two components: a $[\text{2Fe-2S}]$ cluster (orange) $g_{\parallel} = 2.02$, $g_{\perp} = 1.94$, linewidth = 50 G and a radical signal (purple) $g_{\text{iso}} = 2.01$, linewidth = 20 G, 4% relative weight.

Supplementary Data Figure 3e. Raw, as collected EPR spectrum of NqrF-C70A and a background EPR spectrum of a water blank sample, both collected under the same conditions: 10 K; 9.644 GHz; 10 mW microwave power; 7.46 G modulation amplitude; 100 kHz modulation frequency; 81.92 ms conversion time; 20.48 ms time constant. The subtraction of the two spectra is offset vertically for clarity.

The majority of the cluster exhibits a spin state $S=1/2$ and is relaxing unusually fast. We had to use high-power settings to monitor this state. The EPR features of the clusters are

additionally overlaid by a fast-relaxing radical signal present in the sample (please see also our comments on flavin radical species in Na⁺-NQR, reviewer 2, point 11). Moreover, we observed spectral features, which are indicative for a $S=9/2$ state. Such features are observed for clusters with strong geometric strain, like e.g. in variants where one coordinating Cys is replaced by a Ser (see also Reference 3 given in Supplementary Data, Subramanian *et al.*). We include now also further EPR spectra recorded at low and high-power settings of further variants of Na⁺-NQR lacking the cluster in NqrD-E, demonstrating the absence of the $S=9/2$ feature in variants lacking the cluster in NqrD-E (see Supplementary Data Figure 3 c, d).

The samples were reduced with sodium dithionite in the absence of oxygen in a glove box as described in material and methods section (pages 12-13).

The cluster of NqrF has been investigated and described at detail in two previous publications. The two references describing the [2Fe-2S] cluster in NqrF are now included in the revised version of the manuscript.

- Karin Türk, Andrea Puhar, Frank Neese, Eckhard Bill, Günter Fritz, Julia Steuber (2004) NADH oxidation by the Na⁺-translocating NADH:quinone oxidoreductase from *Vibrio cholerae*: functional role of the NqrF subunit. *J Biol Chem* 279(20):21349-55.
- Po-Chi Lin, Andrea Puhar, Karin Türk, Stergios Piligkos, Eckhard Bill, Frank Neese, Julia Steuber (2005) A vertebrate-type ferredoxin domain in the Na⁺-translocating NADH dehydrogenase from *Vibrio cholerae*. *J Biol Chem* 280(24):22560-3

P. 7 – similarly to NqrF above, it should be indicated in which experimental conditions a particular NqrC conformation was observed.

This is now clearly described in the manuscript. In all cryo-EM structures of Na⁺-NQR (determined by us and in a further study by Kishikawa *et al.*) NqrC adopts a conformation where it resides at subunit NqrB. Only in the X-ray structures of Na⁺-NQR the subunit NqrC resides at NqrD-E. The conformation of NqrC observed in the X-ray structures of native Na⁺-NQR is stabilized by a crystal contact. This puts forward that the crystal structure trapped a transient state that is stabilized by this crystal contact (as described in the discussion). If crystals of Na⁺-NQR are treated with the inhibitor DQA subunit NqrC switches from NqrD-E towards

NqrB. This is clearly described in the main part of the manuscript (page 10 lines 1-2, page 10 lines 8-10, page 10 lines 18-20).

P. 6, P.8, Suppl Data Table 2 – it is not clear why the rate of UQ1 reduction is only about 30% or so of the rate of NADH oxidation even in WT? It should be close to 100%. Where do the rest of electrons from NADH go (or leak) to? This should be discussed.

Indeed, the oxidation rate of NADH should correspond to the reduction rate of quinone. The obvious discrepancy between both rates originates from the propensity of Na⁺-NQR to transfer a substantial amount of the electrons from NADH oxidation to oxygen. In particular the FAD located in NqrF contributes to the generation of superoxide. This behaviour has been described previously, and it is now explained in the Materials and Methods section (page 16, line 3-4), where we cite the following references:

- Muras V, Dogaru-Kinn P, Minato Y, Häse CC, Steuber J. (2016) The Na⁺-Translocating NADH:Quinone Oxidoreductase Enhances Oxidative Stress in the Cytoplasm of *Vibrio cholerae*. *J Bacteriol* 198:2307-17
- Lin PC, Türk K, Häse CC, Fritz G, Steuber J. (2007) Quinone reduction by the Na⁺-translocating NADH dehydrogenase promotes extracellular superoxide production in *Vibrio cholerae*. *J Bacteriol.* 189:3902-8

As a note, since we are aware of this behaviour, we are currently establishing an improved assay for Na⁺-NQR with 100% coupling between NADH oxidation and quinone reduction. A publication on this assay is in preparation. We are happy to provide more details if required.

ED Fig. 5d and 9 – the legends should state how voltage was measured here (oxonol).

The figure legend has been changed accordingly.

Fig. 4 – should show also how UQ2 binds, since structures were determined.

We have added a panel in Fig. 4, now Fig 4c, showing UQ2. As stated already in the text, UQ2 binds in the same mode like UQ1.

Modified Figure:

Figure 4. Ubiquinones and inhibitor HQNO share the same binding site in NqrB

a, Ubiquinone-1 (UQ1, green spheres) binds to the rim of NqrB with the head group close to the cytoplasmic aspect of the membrane. **b**, **c** Two amphiphilic helices AH-I and AH-II at the N-terminus of NqrB close upon UQ binding and contribute to the mainly hydrophobic interaction of UQ-1 or UQ-2 with NqrB. The head group of UQ-1 / UQ-2 forms a hydrogen bond to backbone carbonyl of B-Asn156. **d**, Structural alignment of Na⁺-NQR in complex with either UQ-1 or with inhibitor HQNO. Both molecules bind very similarly to the site, however with different positions and orientations of the head groups.

P. 11, l. 2 – why NADH and UQ2 together were used to create reduced state? Just NADH should be enough. Addition of NADH together with UQ2 will create turnover-like conditions – this should be discussed in detail.

It appears that one of the key states (fully reduced by NADH) may be missing from the current structural analysis. The relocation of NqrF cluster towards NqrD-E might be then observed?

To refer to the current state in l. 21 as a “reduced state” seems to be wrong, it probably applies only to NqrD-E cluster.

Yes, indeed using stoichiometric amounts or higher UQ2 concentrations versus NADH concentrations might create turnover-like conditions and the term “reduced” would then be misleading. However, the design of the experiment is different. In the sample there is a 5-fold excess of NADH ($E^{\circ} = -320$ mV) over UQ2 ($E^{\circ} = +90$ mV), i.e. according to the thermodynamics of the reaction all UQ2 will be reduced. That way the reaction cycle runs for a short time and creates reduced UQ2. Once UQ2 is reduced, there is still NADH that will feed further electrons into the electron-transfer chain of Na^+ -NQR. From spectroscopic data from previous studies we know that this procedure will reduce most redox components of NQR. Moreover, by this procedure we ensure that the substrate UQ2 is reduced, too. Thus, the state generated does not represent a “turnover-like” state. A turnover-like” state rather implies that several parts of the electron transfer chain are oxidized while others are reduced.

We agree with reviewer 1 that “reduced state” is misleading since it refers only to the cluster in NqrD-E as pointed out by the reviewer. We have revised this part of the manuscript (page 14) and describe also the redox states of the different co-factors in more detail.

P. 11, l. 22 – experimental structures of oxidised and “reduced”-like state do not differ in NqrC position, so this comment is not clear.

As pointed out by the reviewer the description as “oxidized” and “reduced” is misleading and we have revised this entire part of the manuscript. The wording has been improved to be clearer at this point. In fact, the oxidized state of Na^+ -NQR and the state that we consider to represent a reduced $[\text{2Fe-2S}]_{\text{NqrD-E}}$ cluster differ in the position of NqrC and differ with regard to the conformation of NqrD-E as well as in the interhelical angles of the TM helices of NqrC and NqrF. We revised this part (pages 14-16). We hope with the changes made this part is much clearer now (see also our comment on oxidized versus reduced state of Na^+ -NQR, reviewer 2, point 11).

Reviewer #2:

Hau et al present a series of cryo-EM and X-ray structures of intermediates of the Na-pumping Na⁺-NQR from Vibrio cholerae together with a quite large array of data from complementary techniques to verify/support their findings. The data are of good quality and the topic interesting and novel insights should be possible to extract. However, the manuscript as written needs to be significantly revised, especially in terms of clarifying the resulting model of redox-driven Na⁺-translocation.

We thank reviewer 2 for the positive assessment of the manuscript. We have revised the manuscript as pointed out by reviewer 2 and hope that the changes made clarify the model of redox-driven Na⁺-translocation. We have added a new Figure 6 to the revised manuscript, which depicts a proposed catalytic cycle and added a description of a putative model of redox-driven Na⁺-translocation.

Specific comments in order of when they appear in the manuscript is found below:

1. Introduction to 'state of the art' is very short and gives very little background to what was known and where current work fits in.

In the revised version, a more detailed introduction is provided including background information and unresolved questions in the field (pages 2-4).

2. Also it is hard to understand how much is new in this study compared to the previous X-ray structure, which should be more clearly described.

In the revised version, we address this aspect in more detail in the introduction section (page 4). We describe now the improvement and the new knowledge gained from this study.

3. On page three the previous X-ray structure at 3.5 Å is said to be of low resolution compared to the now obtained cryo-EM structures at 2.1-3.4Å. This seems to be a statement that needs modification and explanation.

This might be indeed misleading and not clear for the broad readership. We include now a short section (page 3) that states, that an X-ray structure of a flexible complex (aggravated by the absence of non-crystallographic symmetry) at 3.5 Å is very challenging to build and refine. Sidechains are not clearly resolved and solvent molecules are not visible in the density. A 3.4 Å cryo-EM map is typically of much higher quality than a corresponding X-ray map. The resolution has been also improved now to 3.2 Å. Most important, at 2.1 Å solvent molecules like water and ions like Na⁺ and K⁺ can be modelled, which is key to resolve a mechanism of Na⁺ pump.

4. For the structure of Na⁺-NQR in complex with NADH, it is unclear why there has been no electron transfer from NADH?

The wording might have been misleading as noted also by reviewer 1. We have revised this section (page 6, lines 7-8). In fact, there has been electron transfer from NADH to UQ-2 in this sample and we can safely assume that most components of the electron transfer chain are reduced. Since the sample contains a large excess of NADH, also reduced NADH is present in the binding pocket. We could confirm this by the high-resolution X-ray structure of the soluble domain of subunit NqrF in complex with NADH.

5. On p.7 The discussion on ET distances is a bit oversimplified. What would be the rate of ET at 22 Å and how much slower is that compared to turnover of the enzyme.

The discussion of ET has been rather short since we tried to put the major conclusion in a nutshell. We revised this part of the manuscript. The electron transfer rate at a distance of 22 Å is extremely slow and not compatible with physiological electron transfer. When applying the Dutton-Moser rule (with a tunnelling barrier $\beta = 1.4 \text{ \AA}^{-1}$ for proteins) (Moser et al, 1992, Nature 355:796-802, see reference 21 in the revised manuscript), a distance of 22 Å and a driving force of ΔG° of -2.9 kJ/mol (ΔE° FMN_{NqrC}/FMN_{NqrB} = 0.06 V, Bogachev *et al.* 2006. Biochemistry 45, 10, 3421–3428) and reorganization energy $\lambda = 1 \text{ eV}$ (λ ranges in proteins from 0.7 to 1.4 eV, Moser *et al.*, 2010, Biochim Biophys Acta. 1797:1573-86, e.g. in flavoprotein old yellow enzyme $\lambda = 1 \text{ eV}$, Kudisch *et al.*, 2020, J Phys Chem B 124:11236-

11249) the electron transfer rate calculates to $2.6 \cdot 10^{-2} \text{ s}^{-1}$. The maximal turnover rate of Na^+ -NQR observed is about 150 s^{-1} (Juarez *et al.* 2009 J. Biol. Chem. 284:8963-72). Thus, at a distance of 22 \AA the calculated electron transfer rate is ca. 5,000 times slower than the enzyme rate.

For a distance of 34 \AA observed in the X-ray structure between the $[\text{2Fe-2S}]_{\text{NqrF}}$ and $[\text{2Fe-2S}]_{\text{NqrD-E}}$ assuming a $\Delta E_0'$ of 0.05 V ($\beta = 1.4$, $\lambda = 1.0 \text{ eV}$) the electron transfer rate calculates to $4 \cdot 10^{-8} \text{ s}^{-1}$, i.e. about 3.7 billion times slower than the enzyme turnover rate.

Such very low electron transfer rates at these large distances make physiological electron transfer rather unlikely. In order to illustrate the effects of the observed conformational changes on the electron transfer, we prepared a table of the calculated electron transfer rates k in Supplementary Data (Suppl, Data Table 4).

6. The large conformational changes that are observed needs to go in a clear model for what their role would be and this model should be presented in the proper manuscript, see below.

In the revised MS, an additional Figure 6 is included showing a scheme of the proposed catalytic cycles and the different the role conformational changes at the different states. The assignment of the different conformations is now discussed in detail on pages 14-15 of the revised manuscript. We have added also an animation of the conformational changes in subunits NqrD-E (Supplemental movie 2). The putative catalytic cycle is now described in the revised manuscript (page 19 bottom to page 22 top). Please note, the model presented is a refined version of our previous model taking now into account the new findings, however, we still lack structural information of all intermediate states depicted in this scheme.

7. UQ-1 and UQ-2 were localized, but what is the native quinol used in the Vibrio cells?(goes back to the lack of introduction)

UQ-8 is naturally present in *Vibrio* membranes and is used by Na^+ -NQR as electron acceptor (see also references cited in the MS: Juarez *et al.* 2012, J. Biol.Chem. 287, 25678 and Steuber *et al.* 2014, Nature 516, 62). This information is now described in more detail in the introduction of the revised manuscript. However, UQ-8 is not soluble in aqueous buffers used for the sample preparation of Na^+ -NQR. Therefore, UQ-1 and UQ-2 with much shorter isoprenoid tails were used.

8. On p.9 the shallow quinol binding site compared to complex I is discussed, but in the case of complex I this feature has been suggested to be involved in the pumping mechanism, so how is this related to the mechanism envisaged here?

This is an interesting question and in fact it is not easy to answer it straightforward. There are studies strongly suggesting that the electron transfer step between FMN_{NqrB} and riboflavin_{NqrB} is involved in Na⁺ translocation (see references and introduction of the revised MS). The data available so far put forward that the quinone just recycles the riboflavin as electron acceptor (see also below, point 11). Thus, quinone reduction might be not involved in Na⁺ pumping.

9. The description of non-competitive inhibition of HQNO is confusing, is it competitive or not? If HQNO exhibits truly non-competitive behaviour, they should be able to bind both at the same time, true here?

Indeed, the mode of inhibition and the structural data appear contradictory, since HQNO has been described as a non-competitive inhibitor for Na⁺-NQR (Biol Pharm Bull. 1999 Oct;22(10):1064-7). Nevertheless, it resembles structurally UQ-1 (Tanimoto coefficient of 0.75) and has been shown to bind to the quinone binding site of NDH-2 and to the quinone binding site of cytochrome *aa3*-600. Thus, one would assume that HQNO binds in Na⁺-NQR also to the quinone binding site. However, as pointed out, this contradicts the observed non-competitive behaviour.

A key to resolve this contradiction might come from detailed studies on cytochrome *b₀₃* oxidase. Cytochrome *b₀₃* oxidase is also inhibited by HQNO in a non-competitive manner (2010, Biochim Biophys Acta 1797:1924–1932 doi.org/10.1016/j.bbabi.2010.04.011). However, later it was found that HQNO overlaps with the quinone binding site. Li et al. describe (PNAS, 2021, 118 (34) e2106750118, <https://doi.org/10.1073/pnas.2106750118>) that in cytochrome *b₀₃* oxidase HQNO displaces only the head group of UQ-8, while UQ-8 remains bound via the hydrophobic tail to the protein.

Our findings indicate that this might apply for Na⁺-NQR, too. The data in the manuscript form now a basis to investigate in future studies in more detail the binding of the inhibitor and UQ.

10. On p.10 it is stated that the authors ‘develop a model of redox-driven Na⁺ translocation’ which is very good. However, the model itself is not presented as a figure in the main manuscript which it absolutely be.

A detailed model of coupling of redox-driven conformational changes and ion translocation is now presented in a new Figure 6. Please see also our answer to comment 6.

11. On p. 11, it’s a bit unclear why the different states observed are assigned to the redox states they are (in general) and why this could not be verified by spectroscopy as the additions are made well in advance of freezing the samples for cryo-EM? Further down, one state is referred to as the ‘reduced’ state in quotation marks. Why could the state obtained not be verified?

The different redox states of the samples have been verified in previous experiments, but this has not been explicitly stated in the submitted manuscript. We have revised this section to get the key points clearer. The term “reduced” was referring mainly to the [2Fe-2S] cluster in NqrD-E. We recognized this is very misleading and revised this section (Assignment of structures to states of catalytic cycle , pages 14-17).

Moreover, with respect to its oxidation / reduction state Na⁺-NQR is quite different from other respiratory complexes. Usually, we assume that respiratory complexes ‘as isolated’ in the presence of oxygen are in a fully oxidized state. However, it has been shown that the redox state of isolated Na⁺-NQR is not a fully oxidized state. Na⁺-NQR produces several flavin radicals during the catalytic cycle and surprisingly contains a stable flavin radical in the ‘as isolated’ state. This flavin radical is assigned to riboflavin. Flavins radicals observed in Na⁺-NQR are now described in the introduction of the revised manuscript, including the following references:

- Juárez, Oscar; Nilges, Mark J.; Gillespie, Portia; Cotton, Jennifer; Barquera, Blanca (2008) Riboflavin is an active redox cofactor in the Na⁺-pumping NADH: quinone oxidoreductase (Na⁺-NQR) from *Vibrio cholerae*. *J. Biol. Chem.* 283 (48): 33162-33167
- Verkhovsky, Michael I.; Bogachev, Alexander V.; Pivtsov, Andrey V.; Bertsova, Yulia V.; Fedin, Matvey V.; Bloch, Dmitry A.; Kulik, Leonid V. (2012) Sodium-dependent movement of covalently bound FMN residue(s) in Na⁺-translocating NADH:quinone oxidoreductase. *Biochemistry* 51 (27): 5414 - 5421

It appears that Na⁺-NQR retains at least one electron in a flavin radical despite the presence of oxygen. The optical spectrum of Na⁺-NQR as isolated shows clearly the presence of oxidized flavin cofactors and of the oxidized [2Fe-2S]_{NqrF} cluster as well as of the [2Fe-2S]_{NqrD-E} cluster. This is verified by EPR and Mössbauer studies in previous studies and in this study. In summary, the one-electron-reduced state of Na⁺-NQR with a stable radical is usually referred to as the ‘oxidized state’. It is important to note, that not all Na⁺-NQR preparations contain the same amount of flavin radical.

Because of this ambiguity of the redox state of Na⁺-NQR ‘as isolated’ we considered each conformation observed in the different structures and assigned these to different states. It has become clear, that we and others have not yet observed all possible states. Therefore, we feel that a careful assignment of the different states is important.

When we introduce the state observed in the crystal structure, we use the term ‘reduced state’ in quotation marks. This is reasonable, although the enzyme preparation used in crystallization is in the ‘as isolated’ state, the conformations of NqrD-E, NqrC and NqrF differ from the conformations of as isolated (‘oxidized’) Na⁺-NQR observed in cryo-EM. As outlined in the text of the original manuscript, the X-ray structure of Na⁺-NQR most likely represents a kinetically trapped state of the catalytic cycle, i.e. the state observed in the X-ray structure does not represent a state at thermodynamic equilibrium. Interprotein contacts in the crystal lattice stabilize this particular conformation of Na⁺-NQR in a catalytically relevant, intermediate state. We therefore assume that the conformation observed in the X-ray structure can be assigned to the reduced state of the cluster in NqrD-E. There remains some uncertainty concerning the redox state of the cluster in NqrD-E. Please note that this cluster is very difficult to monitor spectroscopically and has been overlooked for decades. Here we describe the spectroscopic properties of this unusual cluster for the very first time.

12. The order of results does not make sense. There should be a description of what Na sites etc are seen first and what kind of changes are seen in the different states described before (+/- NADH etc) before trying to compile into a catalytic cycle, and the cycle needs to contain a model for how the changes observed translate into Na⁺ pumping, if the title of ‘redox-driven Na⁺ translocation’ should be relevant.

We agree that a description of the structural and functional properties of the Na⁺-NQR might be confusing. This membrane-bound complex possesses unique cofactors, subunits exposed to

the cytoplasm and periplasm, and exhibits varying conformations. The reviewer suggests that the manuscript should start with the description of the Na⁺ binding site and associated conformations, followed by the analyses of redox cofactors and redox states. However, we have chosen another order which follows logically the subsequent electron transfer reactions catalysed by Na⁺-NQR: After an overview of Na⁺-NQR (Fig. 1) the description starts with NADH oxidation by NqrF (Extended data, Fig. 4). Next, electron transfer to the membrane-bound [2Fe-2S] cluster in NqrD-E which its unique properties (Fig. 2, main text), and to NqrC (Fig. 3) is described. We then report the distinct conformations of NqrC and its movement towards NqrB (Fig. 3, main text). Then, electrons are delivered from FMN (NqrC) to FMN (NqrB) are passed to riboflavin (NqrB) and from there to the substrate ubiquinone which also binds to NqrB (Fig. 4, main text). This last electron transfer step is considered to be linked to Na⁺ transport. The reader now is familiar with the architecture of the Na⁺-NQR, and with the complete electron transfer pathway, and is introduced to Na⁺ (and K⁺) binding sites, and the Na⁺ channel in NqrB (Fig. 5, main text). In this back part of the manuscript we also describe specific binding sites for Na⁺ and K⁺, respectively. This represents important, novel information as K⁺ activates Na⁺-NQR. The findings are summarized in a model of redox-driven Na⁺ translocation (new Fig. 6 in the revised manuscript).

13. On p.14 there is no description of the Na⁺ pathway, only the binding site. What lines the pathway? Is it conserved?

The pathway is mainly lined by backbone carbonyls (B-Leu53, B-Met57, B-Val60, B-Val64, B-Ala67, B-Val161, B-Ile2165, B-Pro269, B-Gly272, B-Glu274, B-Gly334, B-Gly335). Only at cytoplasmic entry site two acidic residues, B-Asp52 and B-Glu157 might be involved. At the periplasmic site the sidechain of B-Glu274 is lining the pathway. All three residues are strictly conserved in NqrB from different species but are only partially conserved in homologous RnfD of the RNF complex. We list now the residues in the revised version of the manuscript (page 19, lines 20-24)

14. The usage of site-directed variants is not properly described. What is the catalytic turnover in the Phe38 and Phe342 variants? The smaller voltage generation is no way is proof enough that effects seen are due to impairment of Na⁺ translocation per se. It needs to be compared to the rate of NADH oxidation and controlled for 'active' enzyme as this can vary extensively

when making mutations. Also the protocol in general for making vesicles seems rather 'harsh' and there should be statements of catalytic rates before and after to verify that the enzyme stays as active after the treatment.

In the revised manuscript the kinetic properties of the NqrB-F338A and NqrB-F342A variants are now reported (see additional panels of Extended Data Fig. 9). Activity was determined at different Na⁺ concentrations. Clearly, as observed for the NADH:quinone oxidoreductase activity of wt Na⁺-NQR, variant NqrB-F342A and variant NqrB-F338A were stimulated by Na⁺(Extended Data Fig. 9). This demonstrates that the introduced mutations in NqrB do not perturb the electron transfer and the structure of the enzyme.

At saturating Na⁺ concentrations the NADH oxidation activity of NqrB-F338A variant is decreased by ca. 5% compared to wt Na⁺-NQR. Please note, the voltage generation / Na⁺ translocation activity of NqrB-F338A variant shrunk by ca. 30%, i.e. a 6-fold stronger decrease than the redox activity. Likewise, the UQ1 reduction rate of variant NqrB-F338A is reduced only by 8% compared to the 30% drop in voltage generation.

For variant NqrB-F342A the NADH oxidation activity is even increased by ca. 7% compared to wt Na⁺-NQR and the UQ1 reduction rate of the variant is identical to wt Na⁺-NQR. In stark contrast, voltage generation is reduced by ca. 30 %.

This information is now given in the revised manuscript (page 20 lines 1-9). We conclude that mutations introduced in the Na⁺ translocation pathway in subunit NqrB did not affect the structural integrity of Na⁺-NQR:

“Interestingly, mutation of either B-Phe338 or B-Phe342 results in 30% lower Na⁺-dependent voltage formation presumably caused by a backflow of Na⁺ during translocation because of an incomplete closure of the gate. The results corroborate that B-Phe338 or B-Phe342 are critically involved in Na⁺ translocation. Noteworthy, electron transfer activity is only marginally affected in both variants (Extended Data Fig. 9) confirming that the structural integrity is not affected by these mutations. ”

The protocol for preparing proteoliposomes has been successfully applied previously with Na⁺-NQR where we did not observe a decrease in catalytic rates of the reconstituted pump (Toulouse, Claussen et al. 2017, reference 36 in the Materials and Methods section).

Proteoliposomes are formed by the gentle detergent dilution method. The method was established for the Na⁺-translocating oxaloacetate decarboxylase by Peter Dimroth (Dimroth 1981, reference 37 in the revised Materials and Methods section). We also applied very brief sonication pulses (20 s) to promote the dispersion of lipids and the formation of small vesicles (reference 38 in the revised Material and Methods part). In the Materials and Methods section of the revised manuscript, we describe the method in more detail, including both references.

15. On page 14-15 the general workings of a redox-driven ion pump is way too much oversimplified. In no way does such a system need to 'set pauses' between electron transfer steps in the way described. The system of course needs to couple ET to uptake/release/transfer of the ion but this can be achieved in many ways, e.g. by controlling the redox potentials such that the ion needs to be moved before ET can occur. This can be done without changing the distance between redox cofactors at all, as exemplified for instance by the cytochrome oxidase proton pump. The section needs to be heavily revised.

Unlike other respiratory redox-driven pumps described so far, the Na⁺-NQR (1) catalyzes electron transfer from the cytoplasmic, to the periplasmic, and back to the cytoplasmic aspect of the membrane, (2) operates a membrane-bound [2Fe-2S] cluster and (3) undergoes large conformational changes during the catalytic cycle, as shown in our study. This is unprecedented for a redox-driven cation pump. Thus, a direct comparison with other redox pumps that have been described at great detail such as cytochrome oxidase, is rather difficult and previous findings on these pumps are not applicable. It should also be noted that Na⁺-NQR, unlike cytochrome oxidase, can operate in a reversible manner (generating a sodium motive force driven by NADH oxidation, or driving NAD reduction by the sodium motive force). Concepts developed for cytochrome oxidase therefore cannot be applied for Na⁺-NQR. Moreover, principles developed previously for redox-driven proton pumps, e.g. modulating the redox potentials of redox cofactors by binding of the coupling cation (H⁺) (protonation/deprotonation) to control electron transfer, fail to explain the processes in Na⁺-NQR, i.e. Na⁺ translocation.

From the structural and functional data available we conclude that Na⁺-NQR assumes definite, intermediate states where the electron transfer is drastically impaired due to the large distance between cofactors, and will only resume once the position of cofactors has changed, again enabling electron transfer. The term "pause" was used figuratively. It is widely used by the

scientific community working on conformational (not necessarily redox) -driven pumps. For example Watanabe et al (Rikiya Watanabe, Ryota Iino, Katsuya Shimabukuro, Masasuke Yoshida & Hiroyuki Noji, 2008, EMBO reports 9, 84–90) use the term “pause” to describe that an intermediate step in the catalytic cycle of F1FO ATP synthase is drastically slowed down in response to a certain conformation of the pump (“stepping rotation mechanism” of ATP synthase). We use the term ‘set pauses’ to illustrate the different time scales and rates of the different steps in the catalytic cycle of Na⁺-NQR. We have included now a table with the calculated electron transfer rates based on their distances in the structures and redox potentials (Suppl. Data Table 4) exemplifying the different time scales of electron transfer versus ion translocation. Please note that the electron transfer rates between two observed conformations e.g. for [2Fe-2S]_{NqrD-E} to FMN_{NqrC} change by factor of $1.1 \cdot 10^{12}$ (i.e. 1,1000 billion-fold) or between FMN_{NqrC} and FMN-NqrB by a factor of $6.9 \cdot 10^9$ (i.e. 6.9 billion-fold). We think that such huge changes in electron transfer are well described by ‘pauses’. Nevertheless, we changed almost all phrases like ‘set pause’ to ‘decelerate’ electron transfer. Moreover, we include now in the revised manuscript a new Figure 6 and text describing a putative reaction cycle of Na⁺-NQR with the different conformational changes connected to steps of ion translocation.

16. In the resulting model (p. 15) which should be outlined in the main text, it should also be made more clear which statements have support in the data and which do not.

See the new paragraph and new Figure 6 where we clearly discriminate between experimentally verified states, and proposed states.

17. On line 18 p15, ‘opening of the gate’ refers to which gate?

We are sorry for this unclear description. We have revised the manuscript and introduce the proposed gate on page 20, lines 2-4, where describe a proposed mechanism of Na⁺-translocation by Na⁺-NQR. The gate refers to the constriction of the Na⁺ translocation pathway by the residues Phe338 and Phe342 in NqrB. We propose that electron transfer in NqrB triggers a transient conformational change that shifts these two residues, which located on a single transmembrane helix, and allows for Na⁺ translocation.

Reviewer #3:

The human pathogen *Vibrio cholerae* relies on the Na⁺-translocating NADH:quinone oxidoreductase (Na⁺-NQR) to generate membrane potential for energy metabolism. It is estimated that Na⁺-NQR accounts for over 80% of Na⁺ pumping activity and voltage formation, making it an attractive target for antibiotic development. The Gunter Fritz lab published a crystallographic study of the *Vibrio cholerae* complex in 2014. This manuscript by Hau et al. expands on and improves the previous study of the same complex. The series of structures of the Na-NQR determined in different ligand binding states highlight conformational changes that are presumably linked to the enzyme functional cycle. They complemented their impressive structural studies with a comprehensive biophysical characterization, demonstrating an intimate coupling of protein conformational changes with sodium translocation and electron transfer process. The authors further augmented their findings with mutagenesis and cross-linking experiments. The structural work is well done, and the mechanistic proposal is interesting and supported by the evidence presented.

Major point

The authors combined spectroscopy and high-resolution structural data to show an imbedded [2Fe-2S] co-factor in the membrane region. However, the presence of such co-factor has been reported in a recent paper by Kishikawa et al. in Nature Communication (2022). The Kishikawa paper also reported conformations similar to those in the current manuscript. They need to cite the literature and compare and discuss when appropriate.

It was not our intention not to cite the other excellent study Kishikawa *et al.* We had submitted our manuscript on July 8th, when the other publication (published July 26th, 2022) had not been out yet. Of course, we are happy to cite the publication by Kishikawa *et al.* and refer to this paper throughout our revised version.

Minor points:

1. Please consider presenting cryo-EM data processing flowchart.

The image processing for all 6 datasets was done in a standard way as described in the Methods section. 2D and 3D classification was used to remove bad particles (see Table 1), but no separate different subsets were defined. Since the procedure was the same for all datasets a

flow chart would contain mainly redundant information. Therefore, we feel that a separate flowchart would not add any further information.

2. Because density modification made a difference, it would be nice to show a side-by-side comparison of before and after densities in key regions.

We added a Figure in the supplement (Supplementary Data Fig. 7) showing side-by-side the effects of map sharpening and density modifications. We have added also a short description how maps were sharpened and resampled in Supplementary Data.

3. Fig. 1d, the distance between the riboflavin and FMN-NqrB is labelled incorrectly. It should be 29.3 Å according to the previous literature.

The riboflavin in NqrB had been placed in the previous study wrongly into a density that originates from DDM. We have placed the riboflavin correctly, therefore the distance has changed.

4. Fig. 3c, the two mutants have the same label. One of them should be L226C/P376C.

Thank you very much for pointing us to this copy/paste error. It has been corrected.

5. Please provide more details in figure legends.

We have revised the figure legends, which should include now more details.

6. Figure 2c: MCD data are usually presented with y-axis in unit of $M^{-1} cm^{-1} T^{-1}$, please also specify the magnetic field strength in the methods section.

Please note, CD data are presented. As described in the method section the CD signal has been calculated from the MCD data which contain the contributions from CD and MCD. MCD spectra have been recorded at a magnetic field of 1.6 T. The details are now given in material and methods (page 13).

7. *The growth curve data shown in Fig. 2g was not discussed in the main text.*

The growth curve is now discussed as well as in the main text of the revised version of the manuscript.

8. *Page 2, Line 6: the word should be “unprecedented”, not “unprecedent”. 9. Page 4 line 24 – “extrapolatation” – check spell.*

Thank you for pointing us to these typos. They have been corrected accordingly.

Decision Letter, first revision:

Message: Our ref: NSMB-A46673A

5th Jun 2023

Dear Dr. Fritz,

Thank you for submitting your revised manuscript "Conformational coupling of redox-driven Na⁺-translocation in *Vibrio cholerae* NADH:quinone oxidoreductase" (NSMB-A46673A). It has now been seen by the original referees and their comments are below. The reviewers find that the paper has improved in revision, and therefore we'll be happy in principle to publish it in Nature Structural & Molecular Biology, pending minor revisions to satisfy the referees' final requests and to comply with our editorial and formatting guidelines.

To facilitate our work at this stage, it is important that we have a copy of the main text as a word file. If you could please send along a word version of this file as soon as possible, we would greatly appreciate it; please make sure to copy the NSMB account (cc'ed above).

Sincerely,

Katarzyna Ciazynska
(she/her)
Associate Editor
Nature Structural & Molecular Biology
<https://orcid.org/0000-0002-9899-2428>

Reviewer #1 (Remarks to the Author):

The manuscript has been revised adequately.

Reviewer #2 (Remarks to the Author):

The revised manuscript by Hau et al. is much improved and I recommend publication.

Reviewer #3 (Remarks to the Author):

The authors have addressed our issues. This is a very nice piece of work that will be appreciated by the NSMB readers.

Final Decision Letter:

Message 17th Aug 2023

:

Dear Dr. Fritz,

We are now happy to accept your revised paper "Conformational coupling of redox-driven Na⁺-translocation in *Vibrio cholerae* NADH:quinone oxidoreductase" for publication as an Article in Nature Structural & Molecular Biology.

As soon as your article is published, you can generate your shareable link by entering the DOI of your article here: http://authors.springernature.com/share. Corresponding authors will also receive an automated email with the shareable link

Your paper will be published online soon after we receive proof corrections and will appear in print in the next available issue. You can find out your date of online publication by contacting the production team shortly after sending your proof corrections. Content is published online weekly on Mondays and Thursdays, and the embargo is set at 16:00 London time (GMT)/11:00 am US Eastern time (EST) on the day of publication. Now is the time to inform your Public Relations or Press Office about your paper, as they might be interested in promoting its publication. This will allow them time to prepare an accurate and satisfactory press release. Include your manuscript tracking number (NSMB-A46673B) and our journal name, which they will need when they contact our press office.

About one week before your paper is published online, we shall be distributing a press release to news organizations worldwide, which may very well include details of your work. We are happy for your institution or funding agency to prepare its own press release, but it must mention the embargo date and Nature Structural & Molecular Biology. If you or your Press Office have any enquiries in the meantime, please contact press@nature.com.

Please note that *Nature Structural & Molecular Biology* is a Transformative Journal (TJ). Authors may publish their research with us through the traditional subscription access route or make their paper immediately open access through payment of an article-processing charge (APC). Authors will not be required to make a final decision about access to their article until it has been accepted. Find out more about Transformative Journals <https://www.springernature.com/gp/open-research/transformative-journals>

Authors may need to take specific actions to achieve [open access](https://www.springernature.com/gp/open-research/funding/policy-)

compliance-faqs"> compliance with funder and institutional open access mandates.

If your research is supported by a funder that requires immediate open access (e.g. according to [Plan S principles](https://www.springernature.com/gp/open-research/plan-s-compliance)) then you should select the gold OA route, and we will direct you to the compliant route where possible. For authors selecting the subscription publication route, the journal's standard licensing terms will need to be accepted, including [self-archiving policies](https://www.springernature.com/gp/open-research/policies/journal-policies). Those licensing terms will supersede any other terms that the author or any third party may assert apply to any version of the manuscript.

Sincerely,

Katarzyna Ciazynska
(she/her)
Associate Editor
Nature Structural & Molecular Biology
<https://orcid.org/0000-0002-9899-2428>
